

**Tectonic interactions during rift linkage: Insights from analog and numerical**
**experiments**
*Timothy Chris Schmid[1], Sascha Brune[2,3], Anne Glerum[2], and Guido Schreurs[1]*
*[1]Institute of Geological Sciences, University of Bern*
*[2]Helmholtz Centre Potsdam – GFZ German Research Centre for Geosciences, Potsdam, Germany*
*[3]University of Potsdam, Potsdam-Golm, Germany*
*Corresponding author Timothy Schmid: timothy.schmid@geo.unibe.ch*
*Institute of Geological Sciences, University of Bern, Baltzerstrasse 1+3, CH-3012 Bern, Switzerland*
*Keywords:* *Numerical modelling, analog modelling, stress deflection, rift interaction, rift*
*propagation*
**Abstract**
Continental rifts evolve by linkage and interaction of adjacent individual segments. As rift
segments propagate, they can cause notable re-orientation of the local stress field so that
stress orientations deviate from the regional trend. In return, this stress re-orientation can feed
back on progressive deformation and may ultimately deflect propagating rift segments in an
unexpected way. Here, we employ numerical and analog experiments of continental rifting to
investigate the interaction between stress re-orientation and segment linkage. Both model
types employ crustal-scale two-layer setups where pre-existing linear heterogeneities are
introduced by mechanical weak seeds. We test various seed configurations to investigate the
effect of i) two competing rift segments that propagate unilaterally, ii) linkage of two opposingly
propagating rift segments, and iii) the combination of these configurations on stress re-
orientation and rift linkage. Both the analog and numerical models show counter-intuitive rift
deflection of two rift segments competing for linkage with an opposingly propagating segment.
The deflection pattern can be explained by means of stress analysis in numerical experiments
where stress re-orientation occurs locally and propagates across the model domain as rift
segments propagate. Major stress re-orientations may occur locally, which means that faults
and rift segment trends do not necessarily align perpendicularly to far-field extension
directions. Our results show that strain localization and stress re-orientation are closely linked,



mutually influence each other and may be an important factor for rift deflection among
competing rift segments as observed in nature.

## 1. Introduction

Continental rifting involves brittle faulting and the formation of subsiding rift basins. In places
where individual rift segments are in proximity, they may interact and link when segments
propagate and the rift system matures (Morley et al., 1990; Nelson et al., 1992; Rosendahl, 1987).
The propagation and linkage of formerly isolated rift segments resembles the propagation and
interaction of extension fractures on a micro-scale (e.g., Childs et al., 1995; Willemse, 1997;
Willemse et al., 1996; Fig. 1a). Indeed, analytical solutions and models have been used to
describe crack growth and to predict its direction (e.g., Macdonald and Fox, 1983; Mills, 1981).
Such cracks occur in a variety of materials over a vast order of magnitude in length scale from
micro-scale cracks in glass to km-scale ridge interaction structures in oceanic crust (Pollard and
Aydin, 1984; Fig. 1a).

Propagation and interaction of individual rift segments occur in continental rift systems at
various scales and have been intensively studied over the years. The East African Rift System
(EARS) constitutes a narrow rift with an eastern and western branch that propagate southward
and northward, respectively (EARS; e.g., Bonini et al., 2005; Bosworth, 1985; Brune et al., 2017;
Corti et al., 2019; Ebinger et al., 2000; Glerum et al., 2020; Heilman et al., 2019; Koehn et al.,
2008; Kolawole et al., 2018; Morley et al., 1990; Nelson et al., 1992). On smaller scale, interaction
of segmented grabens has been studied for example in in the Canyonlands National Park, Utah,
a part of the Basin and Range wide rift (Allken et al., 2013; Schultz-Ela and Walsh, 2002; Trudgill,
2002), where various styles of graben interaction are attributed to the underlying strata (e.g.,
salt layer) or pre-existing weaknesses (Fig. 1b).



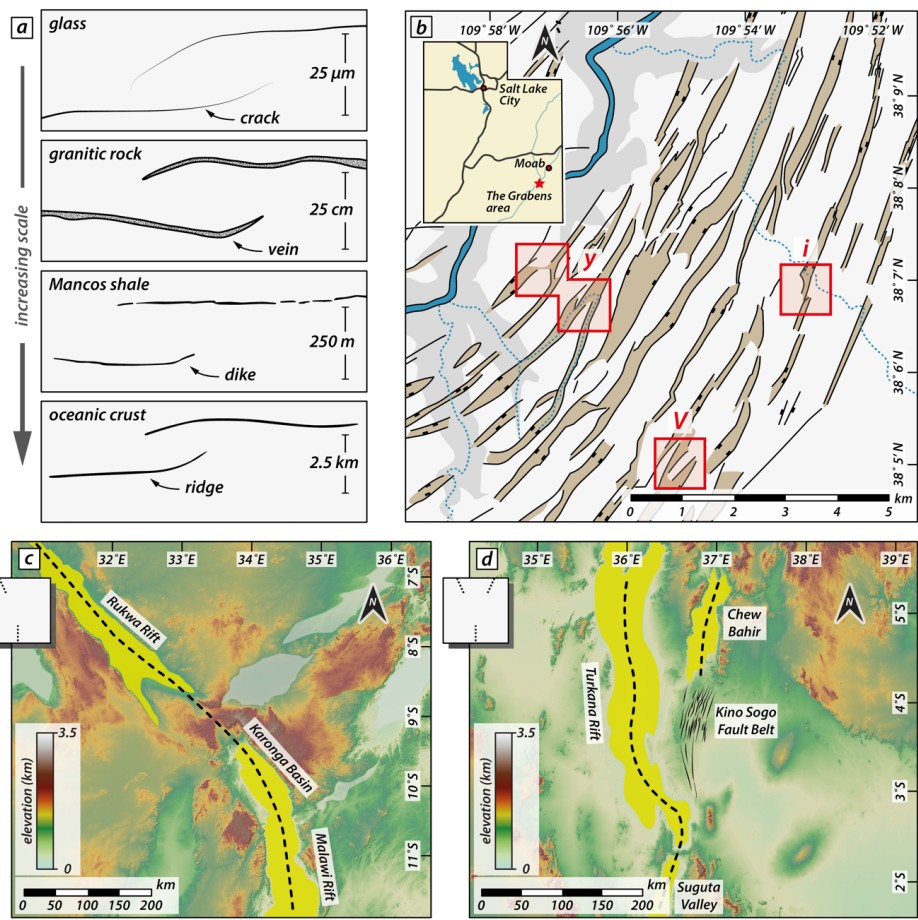

*Figure 1: Similar linkage structures occurring at a vast range of spatial scales. a) Propagation and linkage of segments at different scale from micro cracks in glass to linkage of oceanic ridge segments. Redrawn after Pollard and Aydin (1984). b) Rift-related linked graben structures in the Canyonlands National Park, USA. Red rectangles mark areas with distinct interaction geometries (v-, i-, and y-geometries; see text for detail). Redrawn after Allken et al. (2013). c) Rukwa Rift and North Malawi rift in the western branch of the East African Rift System (EARS). The two basins link obliquely via the Karonga Basin and form an interaction zone. Rift axes redrawn after Kolawole et al. (2021). d) Turkana Rift on the eastern branch of the EARS. The southward propagating Turkana Rift links with the Suguta Valley that propagates northwards. To the east, the Kino Sogo Fault Belt (KSFB) forms the continuation of the Chew Bahir basin which is part of the Kenyan Rift. Rift axes and faults redrawn after Corti et al. (2019) and Vétel et al. (2005), respectively. Grey insets refer to the geometry of the initial pre-existing weaknesses prior to basin evolution (see text for details).*

Structural inheritance is thought to control nucleation and strain distribution along newly formed normal faults as weak fabrics can precondition and weaken a heterogenous upper crust (e.g., Collanega et al., 2018; Heilman et al., 2019; Kolawole et al., 2018; Kolawole et al., 2021;





Morley, 2010; Morley, 1999). Pre-existing weak fabrics may appear as large shear zones (Daly
et al., 1989), suture zones along adjacent basement terranes (Corti, 2012; Corti et al., 2007) or
upper crustal fabrics. Recent strain accommodation in the Rukwa-North Malawi segment of the
western branch of the EARS (Fig. 1c) shows dominant dip-slip faulting parallel to the border
faults (Kolawole et al., 2018; Morley, 2010) driven by the reactivation of pre-existing basement
fabrics (Heilman et al., 2019). There, the concentration of seismicity in the SE and NW of the
Rukwa and Northern Malawi Rift, respectively suggest subsequent propagation and linkage of
the rift segments (Heilman et al., 2019 and references therein).

Rifts may form as initially isolated segments that propagate along strike, interact and evolve
into continuous zones of deformation with time as they link (Nelson et al., 1992). Rift segments
link through previously un-rifted interaction zones resulting in a characteristic geometry that
persists during later rift stages (Nelson et al., 1992). The interaction zone between the Ethiopian
and Kenyan rift of the eastern branch of the EARS comprises different sub-parallel deformed
regions (Fig. 1d). The western rift basin corresponds to the N-S trending Turkana Rift that
propagated northwestward from the Kenyan Rift via the Suguta Valley (Bonini et al., 2005;
Ebinger et al., 2000; Vetel and Le Gall, 2006). The eastern rift corresponds to the Kino Sogo
Fault Belt (KSFB) that propagated southward via the Chew Bahir as part of the Ethiopian Rift
(Ebinger et al., 2000; Moore Jr and Davidson, 1978; Saria et al., 2014). The two branches form
a double-armed system with the KSFB depicting a particular curved faulting style convex to the
west along long fault segments with only minor strain accommodation (Vétel et al., 2005).
However, the reason for the peculiar shape of the KSFB with its characteristic deformation style
remains unclear (Vétel et al., 2005).

Pre-existing fabrics as well as fault interaction across multiple scales disturb the regionally
inferred stress orientation (Morley, 2010; Olivia et al., 2022). In return, stress re-orientations
within and adjacent to rift segments influence the style of progressive deformation. Ultimately,
stress re-orientation may even favor pure dip-slip behavior even for extensional faults with an
oblique orientation to the regional extension (e.g., Corti et al., 2013; Morley, 2010, 2017;
Philippon et al., 2015). This interplay between pre-existing weak fabrics and local re-orientation
of the regional stress field affects how propagating rift segments interact. Under favorable
conditions, it may even cause deflection of propagating rift segments (Nelson et al., 1992).




Rift propagation and segment interaction has been investigated by analog modelling studies
that examined linkage of two segments across a transfer zone (e.g., Acocella et al., 1999;
Bellahsen and Daniel, 2005; Corti, 2012; Zwaan and Schreurs, 2017; Zwaan et al., 2016).
Bellahsen and Daniel (2005) studied the control of existing faults on new fault growth under
multiphase extension. They suggested that pre-existing faults may disturb the local stress field
and impede linkage of newly forming faults. While analog experiments are an effective tool to
simulate mechanical (brittle and ductile) deformation processes occurring in continental rifting
in 3D, accessing information about stresses is challenging. In contrast, numerical modelling
experiments provide direct access to element-wise stress tensors that can be interpreted in
terms of stress regimes and orientations under extension (Brune and Autin, 2013; Duclaux et
al., 2020). Despite the impact of stress distribution on faulting and rift segment interaction, only
recently numerical studies made use of it to gain further insights into rift evolution and
continental break-up (e.g., Brune, 2014; Brune and Autin, 2013; Glerum et al., 2020; Mondy et
al., 2018). However, these studies mostly focus on larger-scale deformation and evaluate
stresses over the entire time span of rifting up to continental break-up.

Here we use crustal-scale analog and numerical models to investigate rift propagation and
strain localization in early rifting stages when rift segments interact. Both types of models
document enigmatic rift segment deflection when two sub-parallel rift segments propagate
approximately in the same direction and compete for linkage with an opposingly propagating
segment. To understand the reason for rift segment deflection, we analyze the stress
distribution in early rifting stages and its interplay with strain localization that initiates above
pre-existing weak fabrics. Our experiments show that relatively simple rift segment interactions
can cause locally complex stress patterns that deviate from the regional stress field. Such stress
re-orientations occur in transient stages and can change over time and with progressive
deformation due to subsequent changes in material strengths.





## 2. Analog model


The presented analog modelling experiment shows unexpected features such as rift deflection.
It motivates our numerical study, and we use the analog model as a reference for examining
strain and stress distribution in numerical experiments.

### 2.1. Analog model setup


For the analog reference model, we use a simplified two-layer crustal scale setup with a brittle
and a viscous material to simulate upper crustal brittle faulting and lower crustal viscous
deformation, respectively. The base of the model consists of a set of alternating plexiglass and
foam bars which are compressed prior to the model preparation by two mobile sidewalls (Fig.
2a). During the experiment the computer-controlled sidewalls extend and provide a symmetric
extension gradient as the model base expands and the model vertically thins. For monitoring
the surface deformation evolution, we use a stereoscopic camera setup to take top view photos
and stereo image pairs every 60 s for quantitative deformation analysis by means of 3D stereo
Digital Image Correlation (Adam et al., 2005). The model was scanned every 20 min in a
medical XRCT scanner for gaining insights on internal model evolution.

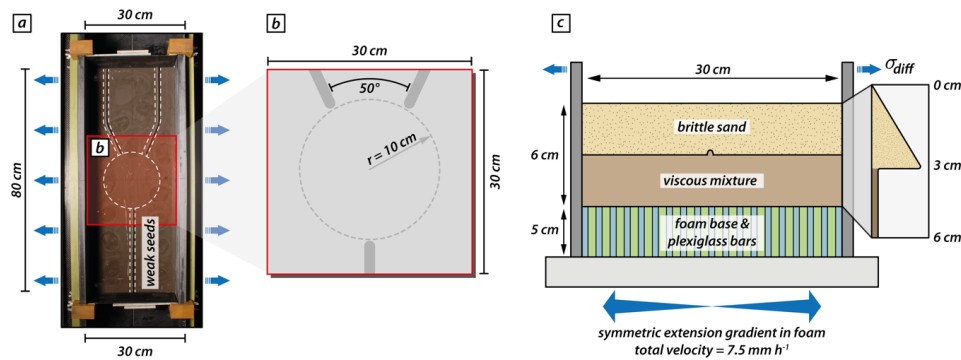


*Figure 2: Analog modelling setup. a) Top view of the experimental apparatus with two mobile side walls that extend*
*orthogonally. The entire model comprises an area of 80 x 30 cm and three viscous seeds are placed on top of the viscous layer*
*before sieving in the brittle sand layer. The central model part where propagating rift segments interact contains no seeds.*
*b) zoom in of the seed configuration into the analyzed model area (i.e., 30 x 30 cm). The two competing seed segments form*
*an intermediate angle of 50°. The model center contains an area with a radius of 10 cm where weak seeds are absent. c)*
*Sketch of the model cross section. The model setup consists of a brittle sand layer representing the upper brittle crust on top*
*of a viscous mixture of PDMS and corundum sand imitating the lower ductile crust.*



### 2.2. Model geometry, rheological layering, and material properties

For simulating upper crustal deformation, we use dry quartz sand with a bulk density of 1560 kg m$^{-3}$ and an internal friction coefficient of 0.72 (Schmid et al., 2020b). For the lower viscous model part we use quasi-Newtonian PDMS/corundum sand mixture (weight ratio 1:1) with a bulk density of 1600 kg m$^{-3}$ and a viscosity of 1x10$^5$ Pa s (Zwaan et al., 2018). Hence, the brittle-viscous setup has a density gradient that avoids density instabilities and spontaneous upwelling of the viscous layer. The model features viscous rods placed on top of the viscous model layer before sieving in the quartz sand. These rods act as mechanically weak seeds and localize faulting in the upper brittle model domain. The used seed configuration includes three individual seed segments. One model side includes a y-seed configuration with one seed segment perpendicular to the extension direction (hereafter called frontal segment) whereas on the opposing side of the model center two obliquely placed seeds (hereafter called rear segments) form an intermediate angle of 50° (see also Fig. 1b&d). The three seed segments hypothetically merge at the model center. However, we exclude weak seeds in an area with a radius r = 10 cm around the model center to allow free interaction of the propagating rift structures (Fig. 2b). The analog model comprises an initial area of 80 cm by 30 cm and has a total thickness of 6 cm (each layer 3 cm) which represents a 30 km thick continental crust. In accordance with the numerical setup, the effectively analyzed model area is restricted to 30 x 30 cm. The mobile sidewalls move with an extension velocity of 5 mm h$^{-1}$ each (totaling in 10 mm h$^{-1}$), which results in a maximum extension of 40 mm at the final model stage after 4h.

### 2.3. Analog model results

In the analog model three different rift segments initiate above the weak seeds and propagate toward each other. Thereby, the two rear segments compete for linkage with the frontal segment. After 30 min (i.e., 5 mm extension; Fig. 3(i)), brittle deformation localizes along two rift boundary faults forming the frontal rift segment. Rifting in the rear segments localizes first along right-dipping rift boundary faults and after 60 min (i.e., 10 mm extension; Fig. 3(ii)) both rear segments develop a set of two conjugate rift boundary faults (Fig. 3a&b (ii)). Interestingly, instead of advancing straight forward, the fault tips deflect and propagate away from each other (Fig. 3b&d(ii)). This is partially due to the rift propagation over the area where no seeds are present where rifting perpendicular to the extension direction is favored. However, after 120





min (i.e., 20 mm extension; Fig. 3 (iii)) rift tips deflect and turn away from one another. Rift tips
deflect from an initially oblique orientation and rotate into an inverted oblique direction (with
respect to the extension direction).

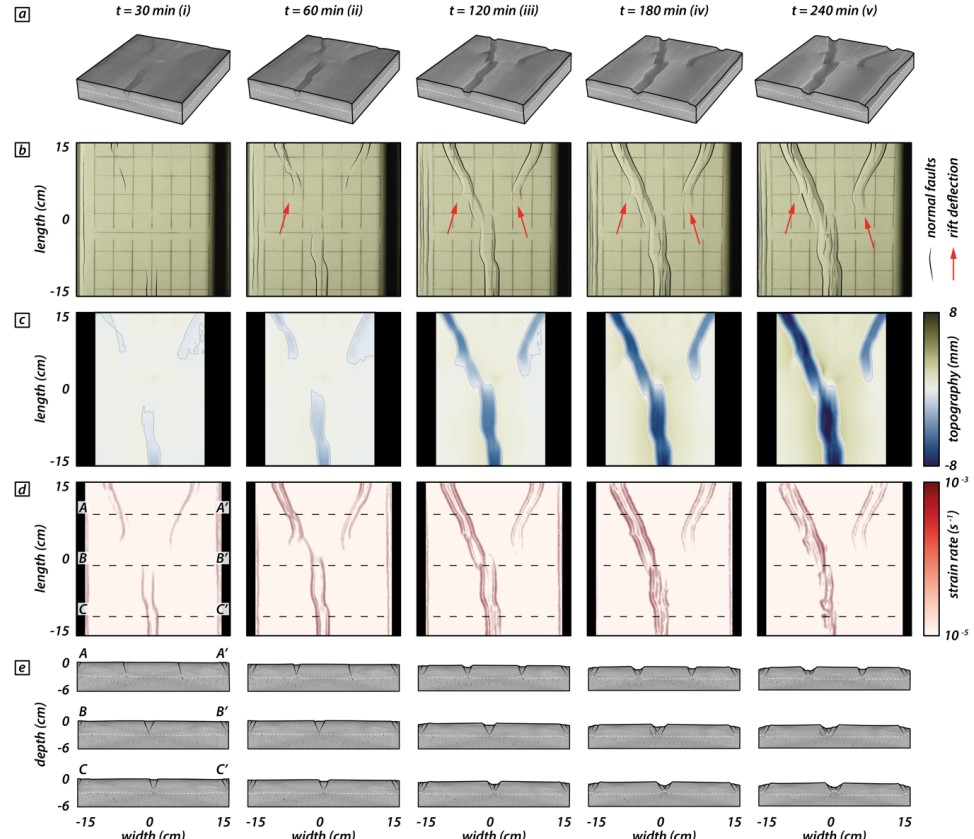

*Figure 3: Analog modelling results documenting deflection of the right rear segment and cessation of faulting activity. Distinct*
*time steps (i.e., after 30 min and after every hour) show the model evolution. a) CT volumes of the investigated model domain*
*at distinct time steps. White dashed lines indicate the brittle-viscous interface. b) Top views and line drawings indicating*
*observable normal faults at the model surface. Red arrows indicate rift tips that deflect and turn away from one another. c)*
*Topography from digital elevation models of the model surface. Colormap from Crameri et al. (2020). d) Strain rates obtained*
*from 3D stereo DIC. Black dashed lines indicate positions of 3 transects through the CT volume. e) Rift transects A-A', B-B',*
*and C-C'. White dashed lines indicate the brittle-viscous interface.*

The frontal and the rear left rift segment propagate further and, as they approach one another,
form an en-echelon basin that convergently overlaps with the frontal rift segment (Morley et al.,



1990; Fig. 3b,d (iii)). After 180 min (i.e., 30 mm extension; Fig. 3(iv)), intra-rift faults develop in the
frontal and left rear rift segments. Note that strain rate is successively localized in the two fully
linked rift segments whereas the right rear segment experiences minor strain rate values (Fig.
3d (iv)). At the final model stage (i.e., after 240 min and 40 mm extension; Fig. 3 (v)), the right
rear segment propagated minimal with a rift tip turned away from the linked segments (Fig.
3b&d (v)). The fully linked frontal and left rear segments continuously accommodated
displacement resulting in deeper rift structures compared to the abandoned right rear segment
(Fig. 3c&e (v)).

## 3.  Numerical modelling
We perform a series of numerical models to investigate rift linkage interaction and to analyze
occurring surface stresses. Similar to the analog experiment, the numerical model consists of a
two-layer crustal setup with laterally homogenous material layers where boundary-orthogonal
extension with constant velocity is applied.

### 3.1. Numerical model setup
We use the open source, finite-element code ASPECT to solve the extended Boussinesq
equations of momentum, mass, and energy in combination with advection equations for each
compositional field (Gassmöller et al., 2018; Glerum et al., 2020; Glerum et al., 2018; Heister et
al., 2017; Kronbichler et al., 2012; Rose et al., 2017). Since the numerical models are motivated
by the analog model, the two set ups are designed in a similar way. To this aim, we employ a
numerical setup where the rheologies of upper and lower crust are brittle and ductile,
respectively, and independent of temperature just like in the analog model. However, the
numerical models operate on the true scales of the continental crust over tens of kilometers
and millions of years, while the analog model is a scaled, cm-sized representation that evolves
on hour-scale. Additionally, the numerical set up applies maximum extension velocities at the
side walls and extension velocities at the base that linearly increase from the center towards
the model boundaries. In contrast, maximum extension velocities at the side walls in the analog
model are achieved via compression of a basal foam plexiglass setup (prior to the model run)
that extends homogeneously during the model run.





The presented numerical experiments cover a rectangular cuboid domain of 150 km width and
length in the horizontal x- and y-direction, respectively, and 30 km in depth along the vertical
z-axis (Fig. 4a). The entire model domain is divided into 1.53 milion hexahedral, second-order
elements. For the upper 15 km of the model, we use a cell resolution of 750 m, with an additional
refinement at the uppermost km which yields near-surface elements with a resolution of 375 m
at the surface. The grid resolution for the lower 15 km of the model is 1500 m. At the left and
right model sides, we apply a symmetrically distributed outflow velocity of ½ $V_x$ = 5 mm yr$^{-1}$,
resulting in a total extension velocity of 10 mm yr$^{-1}$ (Fig. 4a&b). After a total model time of 4 My,
the model has therefore experienced a total extension of 40 km. While $V_x$ is prescribed at the
left and right model sides, $V_y$ and $V_z$ are allowed to move freely. We compensate material loss
through the side boundaries by compensational inflow at the model base and the horizontal $V_x$
component increases linearly from the model center towards the lateral model boundaries (Fig.
4b). The front and back lateral boundaries allow for free slip and the top of the model features
a free surface boundary condition (Rose et al., 2017).

The model includes two rheological layers represented by compositional fields, namely a 15
km thick visco-plastic upper crust with a density of 2700 kg m$^{-3}$ and a 15 km thick iso-viscous
lower crust with a density of 2900 kg m$^{-3}$ and a constant viscosity of 1·10$^{20}$ Pa s. For the upper
crust, the viscous viscosity is fixed to 2·10$^{28}$ Pa s, such that plastic deformation is always
enabled. We introduce initial and dynamic mechanical weaknesses in the upper crust in two
ways. (i) Mechanically weak seeds: At distinct positions near the brittle-ductile interface, the
upper model layer is locally 10% thinned and the lower model layer elevates like the viscous
weak seeds in the analog model setup. These mechanical seeds weaken the upper crustal
strength and localize brittle faulting. Our experiments include three different seed
configurations: v, i, and y (Fig. 4c; see also Fig. 1b-d), where seeds within a central model area
(i.e., r = 100 km) are absent. For each configuration, the rear seeds form an intermediate angle
of 10°, 30°, or 50°. (ii) Friction softening: For each element, an initial plastic strain value of 0
(resulting in strong material) to 0.1 (weaker) is randomly assigned and reduces the maximum
friction angle of 26.56° by a maximum of 10%. This reflects the structural heterogeneity of
natural settings and allows for more randomized strain patterns in the central model domain
where the mechanical seeds are absent. The initial plastic strain noise is distributed over the

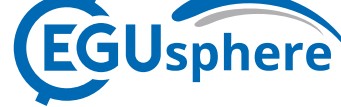

entire model width with an amplitude following a Gaussian curve parallel to the extension
direction that is repeated along the model length (y-direction; Fig. 4d). During continuous
extension, the effective friction angle linearly reduces to 25% of the maximum friction angle (i.e.
to 6.64°) for plastic strain between 0 and 1 while it remains constant at 6.64° for plastic strains
> 1 (Fig. 4e). This corresponds to a reduction of the effective friction coefficient from 0.5 to 0.12.
The cohesion of the upper crust remains constant at $5 \cdot 10^6$ Pa for all conducted experiments.

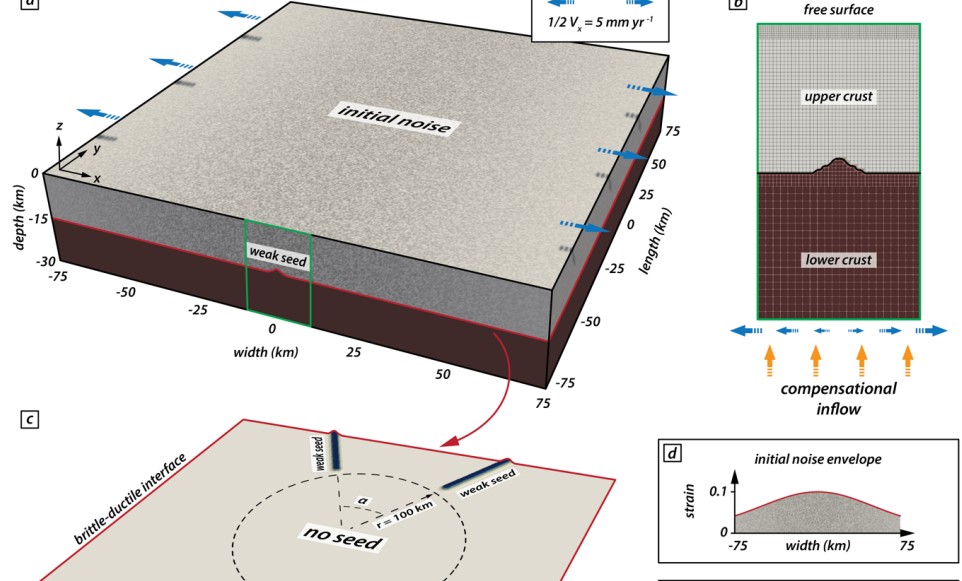

**Figure 4:** *Numerical model setup for iso-viscous models. a) The model domain comprises a volume of 150 x 150 x 30 km. Blue*
*arrows indicate the applied boundary-orthogonal extension. The green rectangle indicates the position of the zoom-in in b).*
*The red line indicates the initial depth of the brittle-ductile interface (as defined by the interface between the two rheological*
*layers) indicated in c). b) Initial conditions and mesh refinement (arrows not to scale). c) Position and configuration of the*
*mechanical weak seeds at the brittle-ductile interface. The setup comprises an area with radius r = 100 km where no weak*
*seeds are present. Three different seed configurations refer to y-, i-, and v-models (see text for details). Colormap from*
*Crameri et al. (2020). d) Initial amplitude of strain along the x-axis. The Gaussian distribution is constant along the y-axis;*
*also see grey shade in a). Note that while the strain amplitude follows a Gaussian distribution, the location of the initial strain*
*is random. e) Linear weakening with strain applied to the friction angle.*



### 3.2. Model limitations
Just like the analog model (Sec. 2), our crustal scale two-layer numerical setup does not
comprise a lithospheric mantle layer and no asthenosphere. Further, the iso-viscous setup does
not account for a temperature-dependent viscosity. However, we focus on an early rifting phase
where the influence of the deforming mantle lithosphere can be neglected. The crustal-scale
setup strongly limits the computational effort for calculating deformation in 3D (Allken et al.,
2011, 2012; Katzman et al., 1995; Zwaan et al., 2016) and hence, our simplifications allow for a
higher model resolution; a necessity to depict early stages of rifting and the coalescence of
brittle deformation. Several alternative model runs have been performed including a
temperature- and pressure-dependent viscosity. Those tests reproduced first-order features
(i.e., strain rates, rift geometry and stress distribution) of the presented models in this study,
which further justified the choice of a simplified iso-viscous setup. Moreover, our model does
not include the influence of melting or magma intrusions nor sedimentation and erosion.

### 3.3. Post-processing
Numerical models pose the advantage that they grant direct access to stress tensors of each
individual cell. We exploit this opportunity by investigating surface stresses to deduct the stress
regime and the effect of different seed configurations on stress distribution. ASPECT provides
post processors that calculate the magnitude and orientation of the maximum horizontal
stresses and the Regime Stress Ratio (RSR) (Glerum et al., 2020). This stress regime
characterization is calculated according to the scheme of the World Stress Map (Zoback, 1992).
The RSR value maps possible stress regimes to an interval between 0 and 3. For isotropic and
homogenous materials, the standard rules of Andersonian faulting are applied (Anderson,
1905). For RSR values < 1, faulting occurs in an extensional stress regime whereas for RSR
values > 2 compressive stress regimes generate thrust faults. Strike-slip faults occur for values
$1 \geq RSR \leq 2$. We extract data of maximum horizontal compressive stress together with the stress
regime and investigate them in areas where the strain rate exceeds a threshold of $10^{-16}$ s$^{-1}$ and
deformation occurs.



### 3.4. General model evolution of the reference model

In this section we describe the numerical modelling results focusing particularly on the general evolution of our reference model with a y-seed configuration and an intermediate seed angle of 50° (Figure 5). At the early stage (i.e., after 0.5 million years), three distinct rift segments develop above the initial seed positions bounded by a pair of conjugate rift boundary faults (Fig. 5a (i)). This early stage is characterized by a symmetric evolution of the two competing rear segments, which results in a symmetric subsidence inside of the graben structures (Fig. 5b (i)). For each rift segment, faulting activity is localized along the rift boundary faults. In the central model domain, however, strain rates depict a more distributed deformation pattern with multiple minor faults (Fig. 5c (i)). Note that the two rear segments propagate and show curved fault segments that initially deflect and turn away from each other resulting in rift segments with a curved geometry expressed in the topography (Fig. 5b (i)). Once they overlap with the propagating frontal segment, faults symmetrically curve inwards and towards the frontal segment. The change from localized strain rates above the seeds to distributed strain rate patterns in the central model domain is best seen in transects (Fig. 5d (i)).

After the first million years, deformation has prominently localized along the left of the two rear segments and along the frontal segment (Fig. 5a&c, (ii)). While deformation in the frontal segment is localized along the rift boundary faults, inward migration occurred in the left rear segment with developing intra-rift faults and only the left-dipping rift boundary fault active. Similarly, the right rear segment shows faulting along the right-dipping rift boundary fault but activity along intra-rift faults is lacking. In the central model domain, formerly distributed deformation localized between the frontal and left rear rift segment (Fig. 5d (ii)). While strain rates indicate a shift from a symmetric to an asymmetric deformation phase, topography is still symmetric which implies that the shift is imminent and has not affected the topography after the first million years (Fig. 5b (ii)).



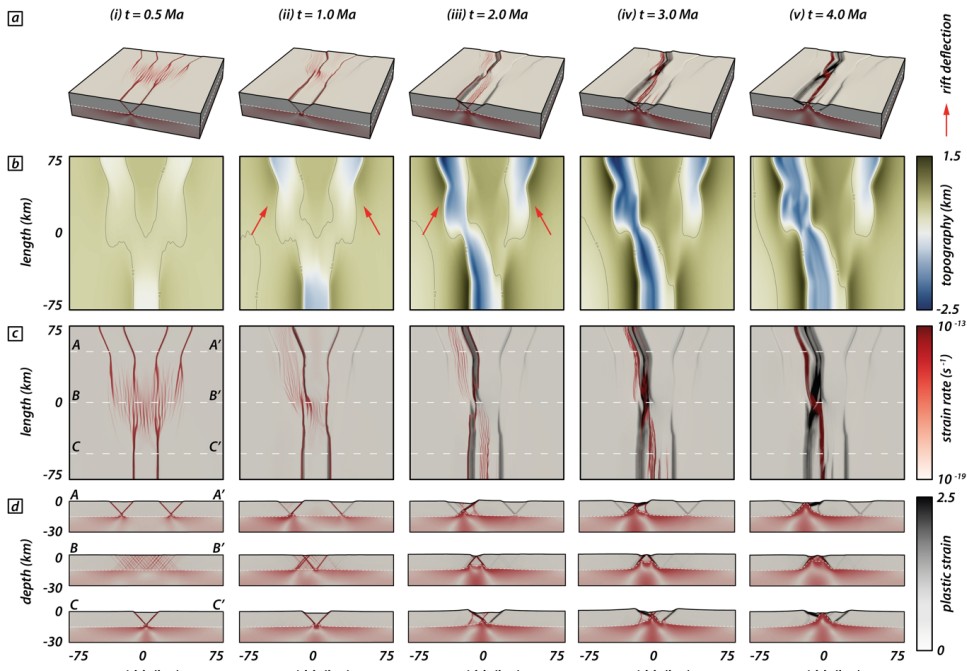

**Figure 5:** *Modelling results of the reference model documenting cessation of faulting activity along the right rear segment while the left rear, and frontal segments link. Distinct time steps show the model evolution. a) Model box showing logarithmic strain rates (red) and plastic strain (black) in the brittle and viscous model domain. White dashed lines indicate the brittle-viscous interface. b) Top views showing the model topography. Red arrows indicate rift tips that deflect and turn away from one another. Black lines refer to the zero elevation height. c) Top views of the model showing strain rates (red) and corresponding plastic strain (black) at distinct model run times. White dashed lines correspond to the three rift transects A-A', B-B', and C-C' in subfigure d). d) Rift-axis perpendicular transects A-A', B-B', and C-C' parallel to the extension direction.*

After two million years, deformation is entirely localized along the frontal and left rear segment. Only the right-dipping rift boundary fault of the frontal segment is active and inward migration led to a set of pervasive intra-rift faults (Fig. 5a&c (iii)). The left rear segment depicts a similar deformation pattern as in the previous step, but strain mainly accumulates along the left-dipping rift boundary fault causing an asymmetric graben geometry (Fig. 5d (iii)). Note that, after two million years, fault activity along the right rear segment completely ceased with no further strain accumulation visible (Fig. 5a,c&d (iii)). The topography reflects this completed switch from a symmetric to an asymmetric deformation stage with enhanced subsidence along the frontal and left rear segments and their linkage throughout the central model domain (Fig. 5b (iii)).






With ongoing extension, deformation subsequently localizes along the axial rift zone that links
the frontal and left rear segments (Fig. 5a,c&d (iv,v)) and faulting activity along rift boundary
faults ceases. The linked structure reaches maximum depth inside of the rift after three million
years. After four million years, however, the basin experiences minor uplift due to increase
upward motion of the underlying viscous material (Fig. 5d (iv,v)). Note that the basin depth of
the right rear rift segment remains stable after two million years and does not experience further
subsidence nor uplift.

**3.5.     Early localization patterns for v-, i-, and y-seeds**
To investigate the influence of different seed configurations, we compare v- (Fig. 6a-c), i- (Fig.
6d-f), and y-seed (Fig. 6g-i) configurations for different intermediate angles (i.e., 10°, 30°, and
50°) at an early stage after 0.5 million years. y- and i-seed configurations provide a setup where
rift structures opposingly propagate towards the model center where rift linkage eventually
occurs. In contrast, rift structures in the v-seed configuration propagate approximately in the
same direction, which has a consequence on the overall strain rate distribution.

The early stage in v-seed experiments (Fig. 6a-c) is characterized by a zone of localized and
distributed deformation in the rear and frontal part of the experiments, respectively. The
transition from localized to distributed deformation occurs where the two competing rift
segments deflect and rotate away from one another. This is consistent with observations for
experiments with a y-seed configuration. However, there the two competing rear segments
rotate back and eventually bend towards the propagating frontal segment (Fig. 6g-i).

For experiment with a i-seed configuration (Fig. 6d-f) two opposingly propagating rift branches
form. Since the right rear segment is absent, both opposingly propagating rift segments link in
the model center where deformation is distributed onto intra-rift faults. The overall strain rate
field is localized, and no strain rate deflection occurs.



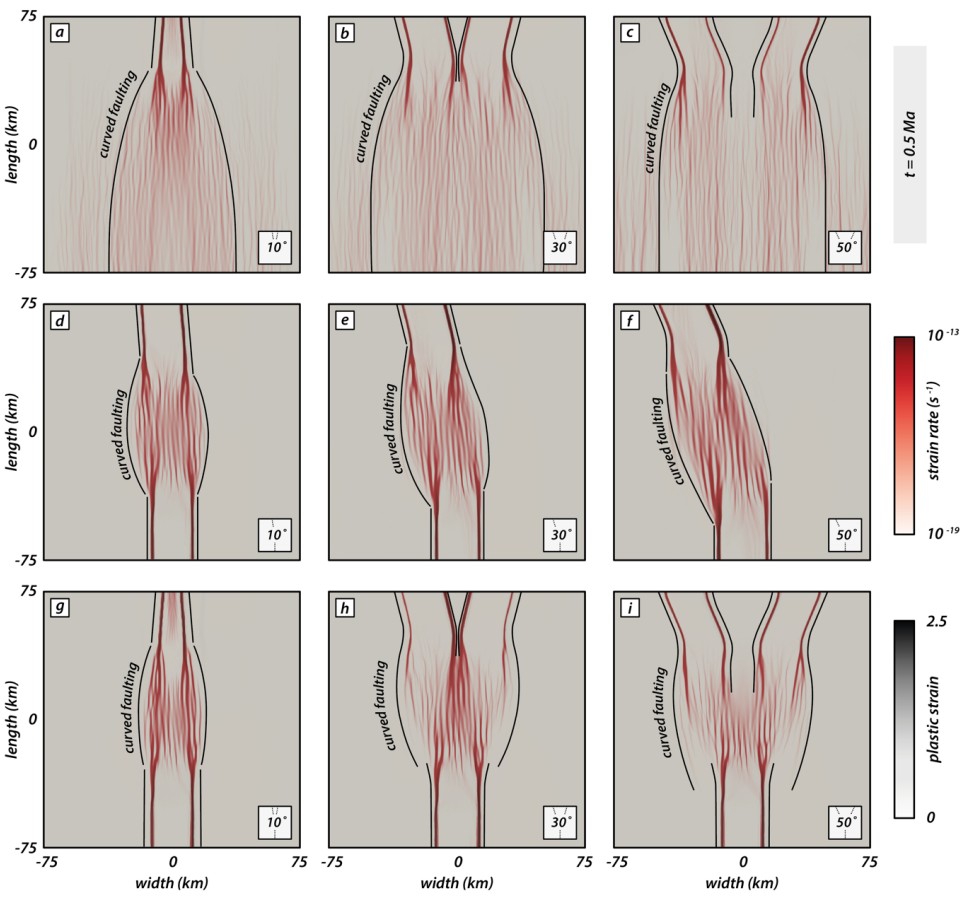

**Figure 6:** *Types of rift segment linkages depending on the seed configuration at an early phase after 0.5 million years. Model top views show strain rates (logarithmic) and plastic strain in red and black colors, respectively. a-c) v-seed configuration for intermediate angles of 10°, 30°, and 50°. d-f) i-seed configuration for intermediate angles of 10°, 30°, and 50°. g-i) y-seed configuration for intermediate angles of 10°, 30°, and 50° (reference model). Black lines confine deformed areas. Curved faulting occurs where rift segments interact.*

Models with a y-seed configuration (Fig. 6g-i) depict a strain rate pattern where deformation is localized along rift boundary faults at the model margins where seeds are present and a distributed en-echelon strain rate pattern in the model center. Note that for the model with an intermediate angle of 10° the two competing rear segments are close enough resulting in a zone where strain is localized along only one rift boundary fault per rift segment (i.e., outward-dipping faults with respect to the model box) that overlap and form a central graben with minor



intra-rift faults. For larger intermediate angles, two individual rift segments (bounded by two rift
boundary faults) form that propagate towards the model center. While the strain rate pattern
due to the competing rear segments is identical for experiments with a y- and v-seed
configuration, the additional frontal segment in experiments with a y-seed configuration causes
localization of strain rates in a single rift branch bounded by two rift boundary faults. This
contrasts with the v-seed configuration where strain rates in the frontal model domain occur
distributed over the entire model domain (Fig. 6a-c).

**3.6.    Final rift geometry and localization patterns for v-, i-, and y-seeds**
The final model stage after four million years best illustrates differences in rift geometry
between the models with different seed geometry and an intermediate angle (Fig. 7). Rift
deflection is well visible in v-seed models (Fig. 7 a-c) and most prominent in experiments with
a larger intermediate angle (Fig. 7b&c). Above the seeds, two short individual rift segments form
bounded by a pair of conjugate rift boundary faults. However, as the rifts propagate towards
the model center, strain is mainly accommodated along the boundary faults that dip towards
the model center. Hence, the larger part of the model subsides uniformly and builds a broad
rift zone confined by two large boundary faults. When the two rift segments propagate, they
deflect and turn away from one another resulting in a gradually wider rift. For intermediate
angles of 30° and 50°, both competing rift segments show active faulting along intra-rift faults
in the rear model part, but a zone of continuous faulting activity has developed along the right
side of the rift.

Models with an i-seed configuration show a continuous and straight rift geometry for all
intermediate angles (fig. 7d-f). For an intermediate angle of 10°, the rift structure is nearly
orthogonal with respect to the extension direction. Note that most plastic strain is
accommodated along the left-dipping rift boundary fault (Fig. 7d). For larger intermediate
angles, the rift subsequently experiences more segmentation with small left stepping segments
towards the rear model part (Fig. 7e&f). Strain accommodation occurs mainly on the right-
dipping rift boundary fault for the frontal model part and switches to the left-dipping boundary
fault in the rear model part.



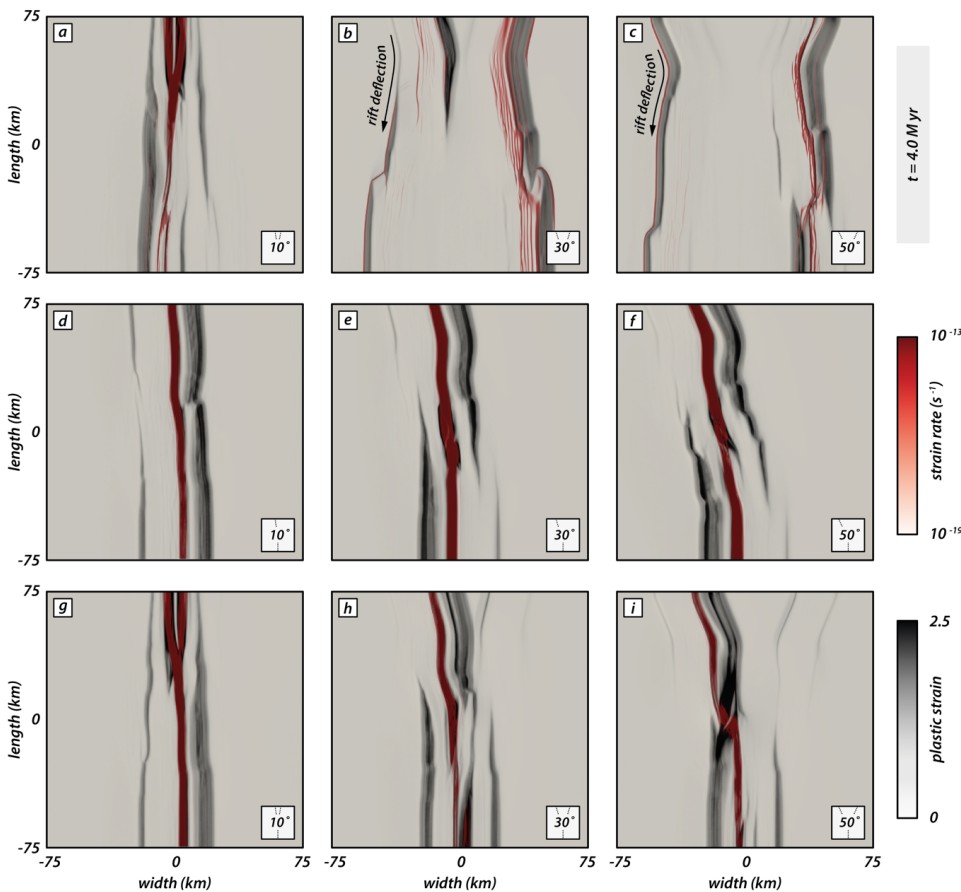

**Figure 7:** *Influence of seed configuration on the final rift geometry after 4 million years. Strain rates (logarithmic) and plastic strain are indicated by red, and black colors, respectively. a-c) v-seed configuration for intermediate angles of 10°, 30°, and 50°. d-f) i-seed configuration for intermediate angles of 10°, 30°, and 50°. g-i) y-seed configuration for intermediate angles of 10°, 30°, and 50°(reference model).*

The most prominent difference occurs in models with a y-seed configuration and various intermediate angles. For an intermediate angle of 10°, the final rift geometry resembles that of a continuous straight rift segment (Fig. 7g). Both competing rear seeds are close enough such that they build one rift system rather than two distinct branches. For y-seed models with a larger intermediate angle (Fig. 7h&i), two individual rear rift segments form and compete for linkage with the frontal rift segment. Plastic strain well illustrates the asymmetric strain accommodation focused along the left-dipping rift boundary fault of the left rear segment, whereas the right



rear segment only experienced minor strain accommodation (Fig. 7h&i). In both cases, high
strain rates are localized in the axial rift zone and witness activity along the linked frontal and
left rear segments.

Note that all experiments with an intermediate angle of 10° (Fig. 7a,d&g) form continuous
straight rift segments, regardless of the seed configuration. Additionally, the final rift geometry
for y- and v-seed configurations for an intermediate angle of 10° is similar with a gently wider
rift in the frontal model part (Fig. 7a&g). In contrast, for i-seed configurations the rift width is
similar along the entire length with a minor lateral offset (Fig. 7d). Strain rates are localized in
the axial rift zone throughout the entire model length forking into two close zones in the rear
end where the competing seeds are located.

**3.7.    S$_{Hmax}$ evolution with progressive deformation**
In this section we present the distribution and orientation of the maximal horizontal
compressive stress component S$_{Hmax}$ with progressive rift evolution and segment linkage. We
focus on models with v-, i-, and y-seed configurations and an intermediate angle of 50° (Fig. 8;
see also supplementary Figures S1-S3) distinguishing between model zones with pre-existing
weak fabrics (i.e., weak seeds) and a central zone where material strength is isotropic.

Our models depict two distinct phases within the first two million years: early strain
accommodation over a wider model domain followed by strain localization and linkage of
propagating rift segments (see also supplementary Figures S4-S6). Consequently, we focus on
S$_{Hmax}$ in the first two million years of deformation and its effect on rift propagation. Figure 8
shows the orientation of S$_{Hmax}$ and the stress regime based on the common color scheme of
the World Stress Map (Heidbach et al., 2018). Note that S$_{Hmax}$ orientation and the stress regime
alone do not suffice to discriminate between locations where stresses exceed crustal strength
and faulting occurs. Strain rate values provide further necessary information, and we use a
threshold of $10^{-16}$ s$^{-1}$ that splits the model into locations of active deformation (i.e., ≥$10^{-16}$ s$^{-1}$) and
tectonically inactive domains (i.e., <$10^{-16}$ s$^{-1}$).








***Figure 8:*** *Interplay of rift localization and surface stresses. Top views show the distribution of the maximum horizontal*
*compressive stress component $S_{Hmax}$ (not scaled to the magnitude) in models with an intermediate angle of 50° at early*
*deformation stages (i.e., until 2 million years). a-e) v-seed configuration. f-j) i-seed configuration. k-o) y-seed configuration.*
*Black colors refer to topographic elevation and red colors mark zones where strain rates exceed a threshold of $10^{-16}$ $s^{-1}$. Color*
*coding for the stress regime marks normal, strike-slip, and thrust faulting in red, green, and blue, respectively, using the*
*common color scheme of the World Stress Map (Heidbach et al., 2018). Elements where the stress regime is non-defined are*
*marked purple. Black arrows highlight stress deflection of $S_{max}$. Rose diagrams show the distribution of $S_{Hmax}$ orientation in*
*zones where active faulting occurs (i.e., strain rate ≥ $10^{-16}$ $s^{-1}$). Large grey arrows for the y-seed configuration mark the change*
*from a symmetric to an asymmetric $S_{Hmax}$ distribution.*

### 491 3.7.1. Effect of $S_{Hmax}$ re-orientation on rift propagation of competing rift segments
### 492 (v-seed models)

Early stages in our numerical experiments are characterized by curved fault traces in the model
center where rift segments interact (see Fig. 6). Hereafter we refer to that phenomenon as
arcuate faulting. Arcuate faulting mainly occurs in experiments with larger intermediate angles
(>10°) in early stages (Fig. 6), especially if two competing rift segments are present (v-, and y-
seed configurations). Moreover, we have shown that deflection of propagating rifts occurs
when deformation is symmetrically distributed along both competing rift branches. This is well
visible for the v-seed configuration (Fig. 8a-e). Assuming orthogonal extension and isotropic
material properties, $S_{Hmax}$ is expected to align perpendicular to the extension direction
producing pure dip-slip normal faults (Anderson, 1905). However, the model shows an
immediate $S_{Hmax}$ re-orientation at early deformation stages (i.e., after 0.4 million years; Fig. 8a)
from a N-S to a E-W orientation in the vicinity of the underlying weak seeds such that dip slip
faults are favored over oblique-slip faults with a strike-slip component. With progressive
extension (Fig. 8b-e), $S_{Hmax}$ re-orientations successively propagate into the isotropic zone
without pre-existing fabrics, concomitant with the rift propagation. Consequently, the position
of the front where stress rotation occurs propagates over time resulting in the deflection of the
propagating rift arms away from each other.

There is a distinct difference between stress deflection along weak fabrics and E-W deflections
of $S_{Hmax}$ in zones where strain rates are below the threshold. The v-seed configuration shows
localized strain accumulation along one rift boundary fault per segment (i.e., the outer one)
resulting in a rift zone with a broad graben system that subsides (Fig. 8e). $S_{Hmax}$ re-orientation



inside of the graben is in parts identical to the E-W orientation of $S_{Hmax}$ outside of the graben.
While local $S_{Hmax}$ rotations may be explained by small differences in the maximum and
intermediate principal stress components, such E-W stress re-orientation in our model occurs
systematically and suggest that this feature reflects the influence of the strength anisotropy
(Morley, 2010). The initial $S_{Hmax}$ deflection near weak fabrics locally favors dip-slip faulting but
also has regional influence on the overall stress regime.

### 3.7.2.    $S_{Hmax}$ evolution in subparallel rift segments (i-seed models)
During the early stage (i.e., after 0.4 million years, Fig. 8f), the distribution of $S_{Hmax}$ resembles
the distribution from the v-seed configuration described in the previous section. Stress
deflection mainly occurs in zones where a weak fabric is present. $S_{Hmax}$ values in the central
zone rotate by a small amount and reflect arcuate faulting (see Fig. 6). Since the two rift
segments propagate in opposing directions, linkage is efficient and localizes in a short time
(Fig. 8f-j). $S_{Hmax}$ values deflect accordingly along propagating faults, which affects the entire
model domain. This deflection does not occur symmetrically on both sides of each rift segment.
Rather, it shows two distinct zones: 1) E-W orientations outside the rift deflect into a parallel
orientation near the rift boarder or 2) N-S orientations outside of the rift deflect into E-W
orientations near faults (Fig. 8j).

We find that $S_{Hmax}$ orientations deflect gradually from E-W to N-S along abandoned rift boundary
faults where activity ceased (Fig. 8h-j; upper left and lower right model domain). In contrast,
$S_{Hmax}$ re-orientations from N-S to nearly E-W towards active rift boundary faults are followed by
a rapid flip back to N-S along the faults (Fig. 8h-j; lower left and upper right model domain). The
two types of re-orientation seem to correspond with two types of deformed zones. Where
deformation is strongly localized along rift boundary faults, jumps in the $S_{Hmax}$ orientation occur.
In contrast, zones where inward migration of fault activity activates intra-rift faults, $S_{Hmax}$ re-
orientation occurs gradually.

### 3.8.    Rift arm competition and deflection (y-seed models)
A prominent feature in our models with two competing rift segments is the deflection of rift
branches and arcuate strain rate patterns (Figs. 8a-e) in the model with a v-seed configuration.
Moreover, the i-seed configuration demonstrates a gradual $S_{Hmax}$ re-orientation over a broader



pre-weakened zone due to formerly active boundary faults. One could therefore expect that
both features should occur in the model with y-seed configuration (Fig. 8k-o).

Indeed, early stages (i.e., after 0.4 million years; Fig. 8k) are characterized by a symmetric stress
field with re-oriented $S_{Hmax}$ values near the two rear rift segments. However, in contrast to the
v-seed configuration, $S_{Hmax}$ re-orientation also occurs near the frontal pre-existing weak fabric
along developing rift boundary faults. In the isotropic zone, $S_{Hmax}$ values dominantly show a N-
S direction. The general N-S orientation reflects the regional stress field due to an E-W
extension as predicted by Anderson (1905) in isotropic areas where rift segments have not yet
propagated into. With ongoing extension, all three rift segments propagate into the isotropic
zone and cause a re-orientation of $S_{Hmax}$ (Fig. 8l). Note that after 0.8 million years the stress re-
orientation occurs symmetrically. This contrasts with the i-seed configuration where $S_{Hmax}$
values deflect either into an E-W orientation along active rift boundary faults or gradually turn
into a fault parallel direction over a broader weakened zone (see subsection 3.7.). The early
symmetric stress distribution in the y-seed configuration model is unarguably due to the
symmetric seed configuration (see also Fig. 8a-e). It is only after 1.2 million years, when fault
activity along the right rear segment ceases that deformation localizes along the left rear and
frontal segments and linkage intensifies (Fig. 8m). Successively, localization and linkage occur
coevally with a switch from a symmetric to an asymmetric stress distribution and resembles
more the stress distribution in the i-seed configuration model (Fig. 8f-j). The model state after
1.2 million years (fig. 8m) also marks the switch from a symmetric to an asymmetric stress
distribution that was formerly dominated by the competing rear rift segments (i.e., v-seed
configuration) whereas after is dominated by the linkage of two obliquely oriented segments
(i.e., i-seed configuration).

This symmetry switch is also visible in rose diagrams of stress orientations within the active
faulting zone (i.e., strain rate $\geq 10^{-16}$ s$^{-1}$). A dominantly N-S oriented $S_{Hmax}$ distribution changes to
a bimodal distribution with a second E-W orientation (Fig. 8l-n). Similarly, bimodal $S_{Hmax}$
distribution is also visible in the experiment with a i-seed configuration but occurs earlier. Since
the experiment with a i-seed configuration is never in the state of an early symmetric stress
distribution linkage is facilitated and occurs earlier (Fig.8g-i).



## 4. Discussion
Despite the relatively simple setup of our experiments, the interaction of individual weak seeds
generates a complex evolution of linkage patterns. In the following we discuss the effect of pre-
existing fabrics on $S_{Hmax}$ re-orientations and how, in return, stress re-orientation influences rift
propagation and rift segment linkage.

### 4.1. Effect of pre-existing fabrics on rift segment propagation, interaction, and $S_{Hmax}$
Previous modelling studies demonstrated that pre-existing weaknesses may cause local re-
orientations of $S_{Hmax}$ resulting in extensional faults with an oblique orientation to the regional
extension direction which exhibit pure dip-slip behavior (e.g., Corti et al., 2013; Morley, 2010,
2017; Philippon et al., 2015). This contrasts the expected (assuming Andersonian faulting theory)
occurrence of faults with an oblique slip component above pre-existing fabrics that are
obliquely oriented with respect to the extension direction (Tron and Brun, 1991; Withjack and
Jamison, 1986). Our $S_{Hmax}$ analysis documents two types of stress re-orientation, either
gradually or by a jump along faults (Fig. 8i). A potential explanation for the two types of stress
deflection is that cessation of boundary fault activity (and subsequent faulting activity along
intra-rift faults) creates a broad zone of reduced crustal strength. Hence, $S_{Hmax}$ orientations
successively re-orient along those formerly active faults and eventually rotate into a N-S
orientation along active intra-rift faults. In contrast, where faulting activity is strongly localized
along rift boundary faults, re-orientation occurs rapidly by a jump from E-W to a N-S orientation.
This suggests that formerly active faults act as a wider zone of pre-weakened material, where
stresses deflect sequentially rather than with a rapid jump. Similar observations have been
made in previous studies of numerical models (Gudmundsson et al., 2010; Kattenhorn et al.,
2000). These experiments suggest that earlier fractures lead to subzones (within a broader
damage zone), where stresses subsequently rotate away from the regional stress field.
Although our analog and numerical models do not feature elastic deformation, they indicate
that stress deflection is an ongoing process, even after elastic material failure. Such a stress
deflection further implies that stress orientations in rocks with pre-existing weaknesses can
substantially deviate from predicted orientations in isotropic media (Anderson (1905)).



It has been proposed that early faulting and propagation in the Rukwa and North Malawi Rifts
(Fig. 1c) were guided by pre-existing basement fabrics (Heilman et al., 2019). This region is
further shaped by a flip in the boundary fault polarity in the present-day geometry within the
interaction zone between Rukwa Rift and North Malawi Rift (Bosworth, 1985). Our i-seed models
show identical geometries for increasing intermediate angles (Fig. 7h&i), where plastic strain
near pre-existing weak fabrics is mostly accommodated along prominent rift boundary faults
that flip fault polarity from the frontal to the rear rift segment. Kolawole et al. (2018) further
propose two different types of strain accommodation at early rift phases. Prominent strain
accommodation localized onto a discrete and narrow zone along large rift boundary faults
(Style-1; sensu Kolawole et al., 2018) and faulting over a broader zone, where fault clusters may
reflect pre-conditioning of the material (Style-2; sensu Kolawole et al., 2018). With this
perspective, jumps and gradual rotation of $S_{Hmax}$ orientations are comparable to Style-1 and
Style-2 strain localization, respectively, as proposed by Kolawole et al. (2018). Hence, the type
of weakness (narrow discrete zone or broad discrete zone) should also be reflected by the
stress re-orientation distribution (Morley, 2010).

**4.2.    Local $S_{Hmax}$ re-orientation and its influence on rift segment interaction and rift**
**deflection**
A particular observation in our experiments with a v-, and y-seed configuration is that two sub-
parallel rift segments, which propagate approximately in the same direction deflect away from
each other at early stages. This is somewhat surprising as one would expect the two rift
segments to cut towards each other by minimizing fault length. The occurrence of rift deflection
in both analog and numerical experiments validates that the results are robust and require
discussing the role of $S_{Hmax}$ re-orientation and how it influences rift segment interaction.

We speculate that, while both rear rift segments in our y-seed models equivalently
accommodate strain in the early stages (i.e., when the overall stress distribution is symmetric;
Fig. 8), $S_{Hmax}$ orientations are dominated by the influence of the two competing rear rift
segments that accommodate strain in equal parts. It is only after fault activity along one rear
segment ceases that deformation localizes along the active rear and frontal segments and
linkage intensifies. Strain localization and linkage occur coevally with a switch from a symmetric
to an asymmetric stress distribution resembling the stress distributions in v-, and i-seed



configuration models, respectively. The switch from a symmetric to an asymmetric stress
distribution in y-seed models also marks the switch from a system that was formerly dominated
by the competing rear rift segments (i.e., v-seed configuration) to a system that is dominated
by the linkage of two obliquely oriented segments (i.e., i-seed configuration).

In models with a v-seed configuration, however, the symmetric phase prevails and causes
coeval $S_{Hmax}$ re-orientations and rift deflection that cause divergence of the two propagating rift
segments. A similar process of extensional segment interaction via stress rotation is known
from mid-ocean ridge settings: Pollard and Aydin (1984) argue that paths of two opposingly
propagating oceanic ridges weakly diverge due to shear stresses that divert propagating ridges
as they approach each other. Once the two ridges overlap, the stress field changes causing
convergence and intersection. Similarly, Nelson et al. (1992) describe interference of
compressional zones of propagating cracks diverting their tips before they overlap and turn
back toward another. In this respect, our models with a v-seed configuration suggest that
stresses also cause divergence of two rift segments that propagate approximately in the same
direction. However, overlap never occurs (as they propagate approximately in the same
direction) and hence, the two segments remain in a stress field that further diverts their paths.

Only in models with a y-seed configuration, compressional zones and rift deflection can be
overcome once the opposingly propagating rift segment links with one of the competing rift
segments. Linkage occurs after about the first million years, concurrently with rift deflection and
abandonment of the right rear segment (Fig. 9b). Moreover, remaining activity in the right rear
segment depicts low strain rates along numerous arcuate intra-rift faults (Fig. 9c). This suggests
that linkage and rift abandonment are closely coupled and faulting along the linked segments
intensifies when the activity along the remaining rift segment ceases. With respect to the
Turkana Region this suggests that the KSFB may represent a southward propagating rift branch
that experienced a limited amount of extension-related deformation before propagation
aborted, similar to the neighboring Ririba Rift in the east (Corti et al., 2019). The definite linkage
of the Ethiopian and Kenyan Rift via the lake Turkana basin may represent a switch from a
symmetric to an asymmetric stress distribution after which local stress re-orientation favored
increased faulting activity along the linked system and caused the abandonment of the young
KSFB as seen in our models (Fig.9). Our modelling results show that stress deflection along rift





segment tips is a mechanical consequence of the interaction between weak zones and far-field
stresses offering a potential explanation for naturally occurring rift deflection such as seen in
the KSFB. However, we must emphasize that complexities in natural rift settings pose additional
difficulties that require further investigations of stress orientations.

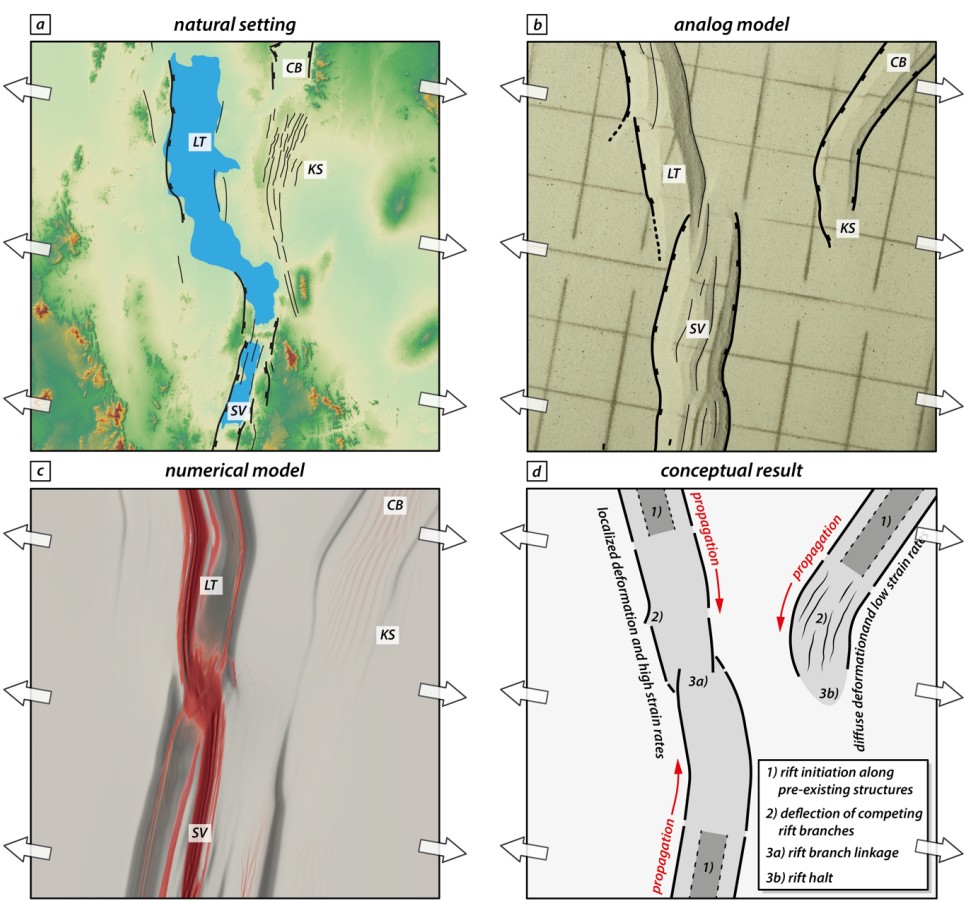


**Figure 9:** *Summary plot showing the geometric similarity of rift segment linkage, deflection of competing branches and*
*abandonment in natural setting, analog, and numerical models. a) Major rift segments in the Lake Turkana Rift system with*
*a double-armed rift system. CB: Chew Bahir basin; LT: Lake Turkana basin; KS: Kino Sogo Fault Belt; SV: Suguta Valley. b)*
*Observed key features at the final stage of the analog model. c) Final strain and strain rate pattern in the numerical reference*
*model. d) Conceptual interpretation of the Lake Turkana Rift based on our numerical results (for details see text).*



Another example of rift deflection has been described in the Main Ethiopian Rift. Geophysical
and geologic studies evidence that pre-existing structures controlled the somewhat 11 Ma
southward propagation of the Northern Main Ethiopian Rift and its contemporaneous westward
deflection along the Yerer-Tullu Wellel Volcanotectonic Lineament (YTVL; Abebe et al., 1998;
Keranen and Klemperer, 2008; Muhabaw et al., 2022). Only after the rotation of the principal
stress direction at about 5-6 Ma (Bonini et al., 2005), extension along the YTVL ceased and
deformation localization along the Central Main Ethiopian Rift became more favorable. Our
models document similar rift deflections and moreover indicate that, even in the absence of
changing plate motions, rift segments deflect, and may cease while competing rift segments
prevail and strain further localizes.

For the Canyonlands National Park, it has been proposed that it is mainly the lateral offset
between pre-existing structures that explains the diversity of structures (Allken et al., 2013; Fig.
1b). With larger offsets, interaction between adjacent rift segments is limited and competing
grabens persist and endure ongoing propagation coevally. We find that stresses, in
combination with the geometry of pre-existing structures, play an important role and that they
have a mutual effect on one another. Hence, stress distribution must be considered as an
important factor especially in early rifting stages when segments link and predetermine strain
localization in following rifting phases.

## 5. Conclusions
We present a series of analog and numerical rifting experiments. Our results suggest that, even
in a relatively simple iso-viscous two-layer crustal setup, pre-existing weaknesses substantially
disturb the regional stress pattern, which impacts rift propagation and the overall rift evolution.
The complex stress re-orientation is distinct for different seed configurations (i.e., v-seed, i-
seed, and y-seed) and closely interacts with the final rift geometry. The most important findings
of our study can be summarized as follows:

•    Our numerical experiments reproduce rift segment deflection seen in our analog
714         models. This highlights the robustness of our results and their applicability to



interpreting rift segment propagation, interaction, and linkage in natural settings of
continental rifting.
• Pre-existing fabrics may control localization of rift segments that successively
propagate into previously undeformed areas. Consequently, stress re-orientation
initially occurs along pre-conditioned zones and propagates, coevally with rift segment
propagation and strain accrual, into formerly undeformed areas.
• Interacting stresses between two competing rift segments may cause outward
deflection of the propagating rift tips resulting in a successively broader rift geometry
along-strike.
• Outward deflection of competing rift segments is less prominent if an opposingly
propagating rift segment is present. With progressive extensional deformation, strain
accrual along one of the competing rift segments prevails whilst faulting activity along
the other segment ceases. Coevally, the general stress orientation changes from a
symmetric to an asymmetric distribution indicating the onset of rift linkage.
• Our modelling results reproduce first-order structures of natural examples from the East
African Rift System and, on smaller scale, graben structures in the Canyonlands National
Park. The combined investigation of surface stresses and strain localization provides an
explanation for distinct rift deflection among competing rift segments and rift linkage
structures where ongoing deformation and stresses mutually affect each other.

While changes in rift orientation are often used to infer regional palaeo-movements, we
demonstrate that local stress field re-orientations can occur under constant plate motions. The
observed stress re-orientations change over time indicating that stresses measured in natural
examples may depict transient stages that change with progressive deformation due to
subsequent changes in material strengths.



## Data availability

Rheological measurements of the used analog materials are available in the form of open access data publications provided by the GFZ Data Service (brittle materials: Schmid et al., 2020a; Schmid et al., 2020b; viscous materials: Zwaan et al., 2018).

## Competing interests

The authors declare that they have no conflict of interest.

## Acknowledgements

We thank Esther Heckenbach for helpful assistance with post processing and visualization. The work was supported by the North-German Supercomputing Alliance (HLRN). We thank the Swiss National Science Foundation for providing financial support.

## Funding

This project is supported by the Swiss National Science Foundation [grant number 200021_178731].

## CRediT authorship contribution statement

Timothy C. Schmid: Conceptualization, Methodology, Investigation, Formal Analysis, Writing – original draft, Visualization, Data curation. Sascha Brune: Conceptualization, Methodology, HPC funding acquisition, Supervision, Project administration, Writing – review & editing. Anne Glerum: Methodology, Software, HPC funding acquisition, Writing – review & editing. Guido Schreurs: Writing – review & editing, Supervision, Project administration, Funding acquisitions, Resources.





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
