# Peer review of "Tectonic interactions during rift linkage: Insights from analog and"

_EGUsphere, 2022_

## Referee Comment (RC2)

Line 76. I suggest adding Morley et al. (2004) JSG. To the references – particulary with reference to lines 77-78 this paper discusses discrete pre-existing fabrics (such as shear zones or fault zones) vs pervasive fabrics, which is what I think you are describing in this passage.

Line 90. I do not understand this passage. Western rift basin? Do you mean western basin in the Turkana area? If so this is wrong, because there are important older rift basins to the west of the rift trend in Northern Lake Turkana (Fig. 1 below). Also the Turkana rift did not propagate northwards from the Kenya Rift. The Turkana area is the site of the oldest rift (Lokichar Basin) in the East African Rift system. The original work on the timing was done in Amoco (myself included), not by Bonini, Ebinger or Vetel and Le Gall.

In Morley et al., 1992 (GSL), 1999 (AAPG studies in Geology 44), Morley (2019, Geosphere), and Morley and Chantraprasert (2022, Ital. J. Geosci., Vol. 141, No. 3 (2022), pp. 295-333,) the models for the evolution of Turkana have changed a bit, but they consistently show the oldest, Paleogene, part of the rift system is in Turkana, possibly down to the Elgayo Escarpment area, and the rift propagated to the south. There is a change from the half graben stage, to the later volcanically dominated graben-in-graben stage (to use an old term), where the boundary fault style is replaced by smaller fault swarms, and the Suguta valley trend and Kino-Sogo Fault belt trend was established. In Turkana this is seen as a shift in fault activity to the east with time.

Line 93. As discussed below it is not cut and dried that the KSFB is part of the Chew Bahr Rift.

Line 98. As discussed below – there are some clear factors we do know are present that make a clear contrast between the Chew Bahr Rift, and the KSFB (timing of activity, influence of basement fabrics vs presence/absence of the Anza Graben and Pliocene volcanics).

Lines 114-115. You might also mention there are some studies where stress deflection has been identified in nature (modern stress from boreholes) deflected around faults (e.g. Tingay et al., 2010, JSG).

Line 498-499 – well visible sounds like a 'street' term. This is clearly visible, or well-display, or well-developed

511 the threshold for what? Failure?

Line 554. Isotropic areas, into which the rift segments have yet to propagate.

Line 558 into either

Line 686 approximately (replace somewhat).

I am not convinced Turkana is configured the way you indicate in Figures 1 or 9. Attached is my fault map from 2019, where you can see that northern Lake Turkana is one trend that continues onshore into the Turkana and Kero basins. This is the older trend and there is no evidence for propagation of this trend to the south (as you show in Fig. 9). Offshore the older history of this trend is difficult to discern, because there is a c. 5 Ma volcanic layer that absorbs a lot of seismic energy so imaging below that layer is poor. But onshore the older part is at least Miocene in age. So it would be propagating to the north if anything. Then we jump to the southern part of the Lake, and that is complicated too. This does seem to be a younger trend in general and it trends a bit more NNE-SSW than the northern trend. In the Loriu and also

in Mount Porr there are older (Oligocene?, Miocene) rift deposits and faults too. But then this younger rift trend is superimposed on them. There does seem to be a NNE-SSW influence of basement trends to the orientation of the Pliocene rift trend. In the southern Loriu this inheritance is quite clear, because basement outcrops with a thin covering of Miocene and Pliocene lavas. What is important about this outcrop – and what is important for your story, is that here we get a rare view that gives us timing. The Miocene lavas are rotated into normal faults. They are unconformably covered by Pliocene lavas, and some faults offset and tilt those Pliocene lavas (Fig. 3). So on these trends you have to very careful about your models and what relatively simple propagation and linkage history you are trying to describe and the actual history of the area. In that Suguta Valley trend, athough it appears to be young, there are actually 3 phases of rifting revealed in the Loriu area – the one that provided the depocentre for arkosic, basement-derived grits (Oligocene or Early Miocene – probably). The one that tilted the Miocene volcanics, and the one that tilted the Pliocene volcanics. If we go to the Mt Porr area there is even an E-W fault trend in basement, whose timing is uncertain (part of the Anza Rift, or part of the EAR?).

The other aspect of the Turkana area that I would like to highlight is the major pre-existing fabric caused by the Cretaceous (reactivated in the Miocene) Anza Graben. It trends WNW-ESE and lies right at the south of the Chew Bahr Rift. I suspect it acted as a barrier to rift propagation. The Chew Bahr Rift shows an unusual rectiliear boundar fault pattern, which is a foliation-fracture pattern in basement.

The Kino-Sogo belt fault pattern is completely different because it is younger than the Chew Bahr Rift, and so instead of the fault pattern being influenced by Precambrian Basement, it is a fault pattern developed on top of both the sediment fill of the Anza Graben, and also Pliocene volcanics that overlie the Anza Graben. So not only is there no basement influence on the fault pattern, there is also the possibility that volcanic processes (dyke intrusion, magma chambers) have influenced the fault pattern. It is not possible for me to categorically say whether the Kino Sogo belt is actually the southwards propagating Chew Bahr rift, or the northwards continuation of the Suguta Valley………….but my prejudice (e.g. Morley et al., 1999) has been that it is part of the Suguta Valley trend (both are of similar age, both involve a volcanic influence. But there is that eastwards step at the north end of south Lake Turkana to explain. Another possibility is that it is it's own independed system and that it propagated both northwards to the Chew Bahr Rift, and south to the Suguta Valley. That might actually make the most sense – that it nucleated around the volcanic centre over the Anza Graben and propagated N and S from there.

Another tricky aspect of the Turkana area is that on the long-term time scale the rift has not propagated in a N-S direction it has actually migrated (apart from the reactivation of some faults in the Anza Graben)

from the west side of the lake to the east-side (Morley et al., 1999, Morley, 2020, Figure 2 below). The oldest part of the rift on a regional scale has actually propagated both to the north and the south from Turkana. The younger part of the rift history that you are focused on in Lake Turkana catches part of the easterly migration – the northern part of the Late being part of the older Miocene trend, with the easterly shift being the Suguta-Kino-Sogo trend. But then as discussed above, even this trend is superimposed on remnants of older rift episodes. In Fig. 4 I made a suggested alternative scenario for Turkana, based on the discussion above.

[Figure]

Figure 2. Topographic map (digital elevation model, slope-shader image Aster Global Digital Elevation Map) of the Turkana region, Kenya, showing key rift features and the location of the study area. Rift structure is from Morley et al. (1999a), with some modification from satellite image interpretation (this study) and from Vétel et al. (2005) and Vetel and Le Gall (2006). FWU—footwall uplift.

Fig. 1.

Morley (2019; Geosphere, v. 16)

[Figure]

Figure 23. Tectonic development of the Turkana–southern Ethiopian Rift area during the late Eocene–recent. Compiled from information in Morley et al. (1999a), Ebinger et al. (2000), Vetel (2005), Vetel and Le Gall (2006), Balestrieri et al. (2016), Emishaw et al. (2017), McDougall and Brown (2009), Brown and Jicha (2016), and Rooney (2017). The Gatome Basin (panel D) is an older basin of probable Late Cretaceous–Paleogene age that underlies the late Eocene–Oligocene volcanic rocks. CB—Chew Bahir Basin; J—Jibisa; LB—Lokichar Basin; OB—Omo Basin; TB—Turkana Basin; TuB—Turkwell Basin. KSFB—Kino Sogo fault belt; SV—Suguta Valley. Pliocene–Holocene volcanic centers (panel A) mapped from satellite images in this study: AV—Asie volcanic centers; DV—Dukana volcanic centers; HV—Hurri Hills volcanic centers; KV—Kulal volcanic hills; MV—Marsabit volcanic centers. Extension direction estimates: 1—Kataboi fault zone area, activity ca. 28–25 Ma (Ragon et al., 2018). 2—Lojamei area; inferred from dip-slip normal faults to be approximately east-west extension; dike orientations are variable and suggest local perturbation of the stress field by magma chambers ca. 17–15 Ma, and more north-south–oriented dikes and approximately east-west extension direction during the 12–10 Ma period of dike emplacement (panels B and C; see the South Lokichar Basin–Lojamei Area section in text). 3—Loriu area; from dike orientations suggesting east-west extension direction (minimum horizontal stress), with strong local stress field perturbation by magma chambers (panel C; see the Loriu Area section in text). 4—Muranachok- Muruangapoi; from fault and dike trends (panel C; see Muranachok-Muruangapoi Area section in text). 5—Napadet-Kamutile-Kathigithigiria Hills; from north-south dike trends, plus local stress field perturbation by magma chambers (panel B; see Napadet-Kamutile-Kathigithigiria Hills section in text). 6—Minimum horizontal stress direction orthogonal to strong NE-SW to NNE-SSW trend of small volcanic cones and craters (see Discussion section) of AV, DV, HV, and MV for the Pleistocene (panel A; Strecker and Bosworth, 1991). Recent extensional activity (red arrows) as indicated by seismicity and deformed Holocene lake shorelines appears to be focused along the trend of Suguta Valley and Lake Turkana (panel A; Pointing et al., 1985; Melnick et al., 2012). Details of Cenozoic activity within the Anza Graben are not shown, but in general, Paleogene activity (panel D) is more important in the central and southeastern part of the graben, while Neogene activity (panels B and C) was significant in the northwestern part (Morley et al., 1999b). When passively subsiding, it was still an important depression and exerted an influence on drainage; during sea-level highstands, it acted as a marine gulf, apparently enabling an unfortunate whale to be preserved amongst freshwater fossils in the Loperot area of the Lokichar Basin, Turkana (Wichura et al., 2015).

Fig. 2.

[Figure]

Fig. 3. South Loriu area. An unpublished figure of mine from a paper on basement inheritance that got rejected Showing the two phases of fault activity, with greater rotation of the Miocene volcanics than the

Plio-Pleistocene ones.

Miocene

Miocene-Pliocene

Chew
Bahr

North Lake
Turkana

?

Kino
Sogo

Pliocene

Anza Graben

Turkana
Basin

Pliocene propagation

Pliocene

Miocene propagation

Suguta
Valley- South
Turkana

Faults initiated during
the Miocene

Faults initiated during the
Pliocene

Fig. 4. A sketch showing what I think are the key components to the fault pattern

---

## Author Comment (AC1)

**Comment on egusphere-2022-1203**

**Title manuscript**

**Tectonic interactions during rift linkage: Insights from analog and numerical experiments**

**Referee's comments in black**

**Authors answers in blue**

**Referee #1 Guillaume Duclaux**

This manuscript investigates the causes of faults deflection during early rift segments propagation. Numerous observations of curved fault systems are reported from early continental rift settings, yet the cause(s) of such deflections remain to be understood. Here the authors use crustal-scale analog and numerical models to investigate rift propagation and strain localisation in early rifting stages when isolated continental rift segments interact. The comparison between nature, analog and numerical models is elegant. Thanks to the high-resolution numerical modelling results the authors demonstrate the importance of transient stress rotations at the surface of early rift systems for controlling propagation of rift arms. Although this work is original and focused on basin scale tectonic evolution of propagating continental rift basins it confirms earlier findings proposed from larger scale numerical models (see point 1 below).

The paper briefly reviews published work on rift basin propagation and linkage, then introduces a set of analog and numerical models' setups designed to investigate rift propagation dynamics in early rifting when separated rift branches interact. The different models explore a wide range of rift arms geometries (imposed using a weak seed heterogeneity at the base of the brittle upper crust) and test how these branches propagate in a sub-pristine environment before joining. Modelling results are presented in great details and highlight transient rift segments deflection prior to propagation. The numerical model's analysis explains this behavior with surface stress rotations and conclude that faults bounding rift segments do not necessary align with the regional stress field. This contribution seems well suited for EGU Solid Earth and will be of interest to the tectonics community in general. Overall, the manuscript is original, very well written, well organized, and beautifully illustrated. I would recommend accepting this manuscript after minor to very moderate revisions.

I present below some key points (mostly related to the numerical models and conclusions) for which I have some concern followed by a list of minor comments and suggestions.

1) My first comment relates to the main conclusion of the manuscript that stress re- orientations occur and change over time and with progressive deformation and the call for caution about paleo-stress measurements @ lines 131-134 & lines 735-739. I couldn't agree more! Yet, I believe that these statements are in essence what we wrote in Duclaux et al. (2020): *"Our models, however, show that progressive deformation during Phase 1 extension results in rotation of the extensional shear zones to become orthogonal to the plate motion direction and control the structural style during oblique rifting. Although the stress around the active extensional shear zones rotates (Fig. 3), the progressive rotation of Phase 1 extensional shear zones during widening (Fig. 5) forces a discrepancy between σ2 direction and the strike of the structures that must be accommodated by a minor component of strike slip. Early rift structures are thus critical in controlling the final architecture of oblique-rifted margins, but because of potential rotations they must be used with caution when interpreting the tectonic evolution of passive margins."*

I hope I'm not biased but I believe a reference to our work here (as the paper is already cited in the MS) would be legitimate, as well as a reference to Gapais et al. (2000) paper. I do understand this original work focus on a smaller scale than ours, but the findings seem to match rather closely.

(Full ref: Gapais, D., Cobbold, P.R., Bourgeois, O., Rouby, D., de Urreiztieta, M., (2000). Tectonic significance of fault-slip data. J. Struct. Geol. 22, 881–888.)

Thank you for pointing this out and the additionally suggested literature. We agree that, albeit on a smaller scale, our conclusions well agree with those in Duclaux et al. (2020) and Gapais et al. (2000) that highlight the issues that arise when interpreting passive margin evolution based on local stress and strain data. We adjusted the abovementioned section in the conclusion and included the suggested references there.

*"Albeit on a smaller scale, implications from our observations are in agreement with conclusions from previous studies (Duclaux et al., 2020; Gapais et al., 2000). Locally, stress and strain can largely deviate from a regional pattern and merely represent local problems of deformation interferences. In addition, the observed stress re-orientations change over time indicating that stresses measured in natural examples may depict transient stages that change with progressive deformation due to subsequent changes in material strengths. This implication must be considered when processing local fault-slip data when interpreting the evolution of rifts at any scale."*

2) Frictional softening is of primary importance to control fault localisation and propagation in the numerical models and I think a few more words should be added about it in the numerical model setup section. Lines 266-268: I understand that grid resolution varies vertically in the top part of the upper crust of the numerical models (Fig 4). Is there a normalization procedure in place for the softening function to account for weakening with different grid sizes (like in Lavier et al., 2000)? If not cells just below the surface will weaken faster than those deeper. That might have rather negligeable effect on the results, but it should be presented/discussed at least briefly.

(Full ref: Lavier, L. L., Buck, W. R., & Poliakov, A. N. (2000). Factors controlling normal fault offset in an ideal brittle layer. Journal of Geophysical Research: Solid Earth, 105(B10), 23431-23442.)

Thank you for pointing this out. We do not use a normalization scheme for frictional softening. As suggested, we added this information in the pertinent section for clarity.

*"… Note that we apply frictional softening as a function of strain within each cell and for simplicity, we do not include normalization accounting for cell size (e.g., Lavier et al., 2000) nor viscoplastic regularization techniques (Duretz et al., 2019; Jacquey and Cacace 2020). …"*

3) Line 328: "the fault segments deflect and turn away from each other": Don't they just tend to form at this angle to strike orthogonal to the extension direction rather than "away from each other" as stated for the analog results (line 192). This brings me to the next point which seems worth discussing further in your work.

This is a good point. We find that, while strain rates tend to strike rather orthogonal to the extension (Fig.5 c), the resultant finite deformation expressed by the topography (Fig. 5b) shows rather curved rift segments that deflect away from each other. To our understanding, this is not in contradiction with results from the analogue model. Fig.3 b and c show the resulting finite deformation (i.e., similar to Fig. 5b) where rift segments deflect from an initially oblique orientation and rotate into an inverted oblique direction (with respect to the extension direction). With this respect we find that analogue and numerical results are rather identical where the rift segments deflect away from each other.

4) Line 504: "dip slip faults are favored over oblique-slip faults with a strike-slip component" - According to Brune et al. (2012) analytical and numerical modelling work oblique extension should be favored. I believe this finding should be discussed in more details as it seems to contradict previous work on the subject. Is this because of the rheologies, the boundary conditions? I find this very interesting.

Thank you for pointing this out. In Brune et al., (2012) rift arms of different obliquity compete with each other and the more oblique one wins (and thus all its secondary features, such as oblique-slip faults). This does not mean that oblique-slip should be favored over dip-slip faulting. The striking difference in the model setup of Brune et al., (2012) and our setup is that ours comprises two sub-parallel rift segments (competing for linkage with an opposingly propagating segment; y- and v-seed configuration) with identical obliquity. Hence, the degree of obliquity should not control which rift segment is favored in our models. Moreover, Brune et al., (2012) and our study involve two different scales (i.e., lithospheric scale vs crustal scale). In our models, the favoring of dip-slip over oblique faults does occur throughout the entire model run but is most prominent at early stages when strain rates are symmetrically distributed and the system is controlled by the competition of the sub-parallel propagating rifts. We therefore conclude that the occurrence of dip-slip fault is largely due to the initial conditions (i.e., the presence of the symmetric seeds in the v- and y-seed configurations).

We agree that this needs some clarification and needs to be discussed in the pertinent section. We adjusted these lines accordingly in section 3.7.3.

*"… The early symmetric stress distribution in the y-seed configuration model is unarguably due to the symmetric seed configuration (see also Fig. 8a-e). At this stage, dip-slip faulting along the competing sub-parallel rift segments is favored over oblique slip faults identical to the v-seed configuration. It is only after 1.2 million years, when fault activity along the right rear segment ceases that deformation localizes along the left rear and frontal segments and linkage intensifies (Fig. 8m). Successively, localization and linkage occur coevally with a switch from a symmetric to an asymmetric stress distribution and resembles more the stress distribution in the i-seed configuration model (Fig. 8f-j). The model state after 1.2 million years (Fig. 8m) also marks the switch from a symmetric to an asymmetric stress distribution that was formerly dominated by the competing rear rift segments with dip-slip faulting favored along the two competing rift segments (see also v-seed configuration; Fig.8 a-e). After 1.2 million years the system is dominated by the linkage of two obliquely oriented segments (i.e., i-seed configuration). Note that after 1.2 million years dip-slip faulting mostly occurs along the competing rift segment that links with the opposingly propagating segment whereas dominantly oblique slip faults occur along the abandoned rift segment where activity ceases.
…"*

5) I find Figure 8 very informative and pretty well designed. It allows visualizing stress deflection at the surface of the models and the surface stress regime at once. There must be an interpolation method used for the stress vectors representation as not all stress markers (one per cell cell) are depicted. Could you comment on this and how does it smooth the signal out?

Thank you! The stress vectors are indeed resampled. For this, we defined an equidistant grid plane in Paraview with the desired grid resolution and resampled the existing unstructured stress data on that structured grid plane.

More importantly, I have some trouble with the location marked with "rotation jump" in Fig 8i. It seems that some of the stress markers are not resolved (non-defined in the caption), so I assume $S_{Hmax}$ could be as depicted or be orthogonal?? How can a jump be argued in this context? I'm not arguing it doesn't take place, only that the marked region chose to highlight it is not the most suitable one... It seems to me that the overall "rotation jump" is related to the transition from compressional to extensional regions, while the gradual rotation relates to region with a transition from strike-slip to extensional. Is that correct?

The choice where to show a rotation jump is unfortunate from our side. At its current state, it appears as if "rotation jump" relates to the flip of E-W striking $S_{Hmax}$ (compressional regime/blue) to the N-S striking $S_{Hmax}$ (non-defined regime/purple). However, we refer to the rapid switch from compressional to extensional regions that is associated with rift boundary faults that prevail tectonic activity over long period (i.e., no rift-inward migration) in contrast to zones where this re-orientation occurs gradually via faults with an oblique-slip in a strike-slip regime, as assumed correctly. We adjusted the position of the label and marker in Fig.8 accordingly.

6) The comparison of rift arms propagation, symmetry, and timing between the different model geometries in the discussion would benefit some additional words related to the consequences of differential frictional softening rates resulting from the different seeds geometries. Rather than comparing time between models, you could maybe compare the amount of extension accommodated at one seed tip?

+ Lines 557-563: Because strain is distributed in the 2 arms in the early stage of the v- and y-models, this difference with the i-model is to be expected. Indeed, the models don't have comparable strain rates, and frictional softening isn't as effective in the y- and v- models.

+ Lines 575-576: In the i-model case, frictional strain softening rate is more effective too.

We agree that differences between models are also due to different frictional softening rates in particular branches. But since these differences arise from localisation and competition, we feel that it would not add significant information to compare rift tips at similar total strains.

**Minor comments**

+ Figure 1: a location map for c and d in the context of the EARS would be a nice addition.

We adjusted Fig. 1 and added an overview map of the EARS. This overview replaces the example of the Turkana Rift system since this example of a natural y-configuration has a more complex evolution than we originally described.

+ line 113: for the multiphase extension I would recommend adding a citation to Duffy et al. (2015) whose work seems relevant in this context.

(Full ref: Duffy, O.B., Bell, R.E., Jackson, C.A.-L., Whipp, P.S. & Gawthorpe, R.L. 2015. Fault growth and interactions in a multiphase rift fault network: Horda Platform, Norwegian North Sea. Journal of Structural Geology, 80, 99–119. DOI: 10.1016/j.jsg.2015.08.015)

Thank you for the hint. The study's conclusions are indeed similar to those of Bellahsen et al. (2005) and demonstrate that observations from the laboratory are in agreement with results from natural examples. We have changed this passage accordingly.

*"They suggested that pre-existing faults may disturb the local stress field and impede linkage of newly forming faults which also occurs in natural examples of multiphase extension (Duffy et al., 2015)."*

+ line 213: "[...] propagated minimalLY [...]" - missing the LY

Thank you for pointing this out. We changed it accordingly.

+ Figure 5: Just a question: did you try a model without the random seeds to check whether surface ruptures remain symmetrical? I understand the random noise distribution will promote dissymmetry; this is out of curiosity.

Yes, during the testing phase of the model setup we switched the initial plastic strain distribution on/off. In both configurations, eventually linkage of one of the competing rift segments with the opposingly propagating rift segment is favored. However, we find that the presence of initial plastic strain yields more natural (i.e., distinct) fault zones. This is particularly the case in early phases (i.e., up to 0.5 M years; see image below).

No initial plastic strain

Initial plastic strain ([0, 0.1])

Time: 0.1 M yr

Time: 0.1 M yr

Time: 0.5 M yr

Time: 0.5 M yr

Time: 1.0 M yr

Time: 1.0 M yr

Time: 1.5 M yr

Time: 1.5 M yr

Time: 2.0 M yr

Time: 2.0 M yr

[Figure]

+ Figure 6: While I can understand the "Curved faulting" contour line for i- and y- geometries, I struggle with v- geometries... there are plenty of faults outside the curved faulting region... and the faults within the regions do not seems to be very curved either.

We agree that "curved faulting" is not an appropriate term for the v-seed configuration. Rather, faults deflect in a fan-shape fashion and successively rotate into an orthogonal orientation (with respect to the extension direction) towards the model margin. We therefore use the term "deflection" in the revised version for the v-seed models and adjusted the text where needed. We also adapted this change in Fig. 6 and the pertinent caption.

+ Figure 9: In the models, main border faults are facing each other's creating a strong asymmetry of the segments at time of propagation/linkage. On the other hand, in the natural example LT is marked as a hemi-graben with east dipping western border fault, but SV is super narrow graben and doesn't display apparent asymmetry. Can you please comment on this significant difference?

As commented earlier, we removed the comparison with the Turkana Rift from the revised manuscript since it has undergone a more complex evolution compared to our models. However, we agree that the asymmetry (i.e., half graben vs narrow graben) in models with a y-seed configuration is an interesting aspect that needs further discussion. This feature also occurs at times in models with an i-seed configuration, where most of the strain is accommodated along one prominent boundary fault with a polarity switch across the interaction zone. We implemented this point in the pertinent sections in the discussion.

+ Line 131, 581: I would recommend using the term "heterogeneity" or "structure" rather than "fabrics" throughout the manuscript, but this is just semantic, and I will let the authors decide whether this is the correct terminology. To me a "fabric" relates to a preferred orientation or configuration of all the elements that make up a rock. In the context of this study there is no initial fabric in this sense, but a pre-existing weak structure at the base of the brittle upper crust. A "fabric" would relate to the initial noise distribution within the upper crust region.

Thank you for elaborating this fine difference. We agree and replaced "fabrics" by "structures", where appropriate. Note that, where related literature is cited, we still use "fabrics" according to the literature.

+ Line 621: I'm a little confused... how can a "discrete zone" be "broad"? Maybe the broad zone could be described as "distributed"?

We agree that this reads confusingly. The broad zone (according to Kolawole et al., 2018) describes a wider zone where faulting occurs clustered (rather than along a single discrete fault) and thus, creates a "wider zone" where faulting occurs distributed. We rephrased for clarity:

*"Prominent strain accommodation localized onto a discrete and narrow zone along large rift boundary faults (Style-1; sensu Kolawole et al., 2018) and faulting distributed over a broader zone, where fault clusters may reflect pre-conditioning of the material (Style-2; sensu Kolawole et al., 2018)."*

---

## Author Comment (AC2)

**Comment on egusphere-2022-1203**

**Title manuscript**

**Tectonic interactions during rift linkage: Insights from analog and numerical experiments**

**Referee's comments in black**
Authors answers in blue

**Referee #2 Chris Morley**

The paper by Schmid et al. presents analogue and numerical modelling of the rift segments, to investigate their propagation and interaction, and to understand how stress patterns are affected during propagation and interaction. The paper is well written and illustrated and will be a very good and useful addition to the literature on this subject. I have some minor comments, and one significant issue with the manuscript that are discussed in the attached word document. The significant issue is the way the deformation in Turkana is described - and in the attached document is describe the way I see the development of Turkana. Perhaps some examples form largely non-magmatic rifts outside of the EAR would be useful, since the authors note that the modelling does not consider the effects of magma on rift segment interaction.

I am not convinced Turkana is configured the way you indicate in Figures 1 or 9. Attached is my fault map from 2019, where you can see that northern Lake Turkana is one trend that continues onshore into the Turkana and Kero basins. This is the older trend and there is no evidence for propagation of this trend to the south (as you show in Fig. 9). Offshore the older history of this trend is difficult to discern, because there is a c. 5 Ma volcanic layer that absorbs a lot of seismic energy so imaging below that layer is poor. But onshore the older part is at least Miocene in age. So, it would be propagating to the north if anything. Then we jump to the southern part of the Lake, and that is complicated too. This does seem to be a younger trend in general and it trends a bit more NNE-SSW than the northern trend. In the Loriu and also in Mount Porr there are older (Oligocene(?), Miocene) rift deposits and faults too. But then this younger rift trend is superimposed on them. There does seem to be a NNE-SSW influence of basement trends to the orientation of the Pliocene rift trend. In the southern Loriu this inheritance is quite clear, because basement outcrops with a thin covering of Miocene and Pliocene lavas. What is important about this outcrop – and what is important for your story, is that here we get a rare view that gives us timing. The Miocene lavas are rotated into normal faults. They are unconformably covered by Pliocene lavas, and some faults offset and tilt those Pliocene lavas (Fig. 3). So, on these trends you have to very careful about your models and what relatively simple propagation and linkage history you are trying to describe and the actual history of the area. In that Suguta Valley trend, although it appears to be young, there are actually 3 phases of rifting revealed in the Loriu area – the one that provided the depocenter for arkosic, basement- derived grits (Oligocene or Early Miocene – probably). The one that tilted the Miocene volcanics, and the one that tilted the Pliocene volcanics. If we go to the Mt Porr area there is even an E-W fault trend in basement, whose timing is uncertain (part of the Anza Rift, or part of the EAR?).

The other aspect of the Turkana area that I would like to highlight is the major pre-existing fabric caused by the Cretaceous (reactivated in the Miocene) Anza Graben. It trends WNW-ESE and lies right at the south of the Chew Bahr Rift. I suspect it acted as a barrier to rift propagation. The Chew Bahr

Rift shows an unusual rectilinear boundary fault pattern, which is a foliation-fracture pattern in basement.

The Kino-Sogo belt fault pattern is completely different because it is younger than the Chew Bahr Rift, and so instead of the fault pattern being influenced by Precambrian Basement, it is a fault pattern developed on top of both the sediment fill of the Anza Graben, and also Pliocene volcanics that overlie the Anza Graben. So not only is there no basement influence on the fault pattern, there is also the possibility that volcanic processes (dyke intrusion, magma chambers) have influenced the fault pattern. It is not possible for me to categorically say whether the Kino Sogo belt is actually the southwards propagating Chew Bahr rift, or the northwards continuation of the Suguta Valley............but my prejudice (e.g., Morley et al., 1999) has been that it is part of the Suguta Valley trend (both are of similar age, both involve a volcanic influence. But there is that eastwards step at the north end of south Lake Turkana to explain. Another possibility is that it is its own independent system and that it propagated both northwards to the Chew Bahr Rift, and south to the Suguta Valley. That might actually make the most sense – that it nucleated around the volcanic center over the Anza Graben and propagated N and S from there.

Another tricky aspect of the Turkana area is that on the long-term time scale the rift has not propagated in a N-S direction it has actually migrated (apart from the reactivation of some faults in the Anza Graben) from the west side of the lake to the east-side (Morley et al., 1999, Morley, 2020, Figure 2 below). The oldest part of the rift on a regional scale has actually propagated both to the north and the south from Turkana. The younger part of the rift history that you are focused on in Lake Turkana catches part of the easterly migration – the northern part of the Late being part of the older Miocene trend, with the easterly shift being the Suguta-Kino-Sogo trend. But then as discussed above, even this trend is superimposed on remnants of older rift episodes. In Fig. 4 I made a suggested alternative scenario for Turkana, based on the discussion above.

Thank you very much for this detailed explanation which gives detailed insights into the evolution of the Turkana area. We agree that the Turkana Rift has undergone a much more complex evolution than how we described it. The present-day geometry of the Turkana Rift tempted us to use it as an example of rift deflection and rift cessation (with respect to the Kino Sogo Fault Belt) if two sub-parallel rift segments compete for linkage with an opposingly propagating segment. Therefore, we excluded the Turkana Rift in the revised manuscript as a natural example of a y-configuration. However, the revised manuscript still adheres to the original interpretation of our y-seed models, where influences from the i-seed, and v-seed configurations lead to two distinct phases of rift evolution (i.e., first symmetric phase, dominated by the competition of the sub-parallel rift segments, and a second phase after linkage where strain localizes along the linked segment and tectonic activity ceases along the remaining segment).

**Minor points**

Line 76. I suggest adding Morley et al. (2004) JSG. To the references – particularly with reference to lines 77-78 this paper discusses discrete pre-existing fabrics (such as shear zones or fault zones) vs pervasive fabrics, which is what I think you are describing in this passage.

Thank you for the additional study. Indeed, Morley et al. (2004) fits well here and is an important contribution regarding structural inheritance. We added the suggested reference at the suited line. Moreover, Morley et al. (2004) discusses the issue of local stress rotations which complicates the interpretation of regional tectonics. We also added this reference with respect to that matter in the conclusions.

Line 90. I do not understand this passage. Western rift basin? Do you mean western basin in the Turkana area? If so, this is wrong, because there are important older rift basins to the west of the rift trend in Northern Lake Turkana (Fig. 1 below). Also, the Turkana rift did not propagate northwards from the Kenya Rift. The Turkana area is the site of the oldest rift (Lokichar Basin) in the East African Rift system. The original work on the timing was done in Amoco (myself included), not by Bonini, Ebinger or Vetel and Le Gall.

In Morley et al., 1992 (GSL), 1999 (AAPG studies in Geology 44), Morley (2019, Geosphere), and Morley and Chantraprasert (2022, Ital. J. Geosci., Vol. 141, No. 3 (2022), pp. 295-333,) the models for the evolution of Turkana have changed a bit, but they consistently show the oldest, Paleogene, part of the rift system is in Turkana, possibly down to the Elgayo Escarpment area, and the rift propagated to the south. There is a change from the half graben stage, to the later volcanically dominated graben-in-graben stage (to use an old term), where the boundary fault style is replaced by smaller fault swarms, and the Suguta valley trend and Kino- Sogo Fault belt trend was established. In Turkana this is seen as a shift in fault activity to the east with time.

Line 93. As discussed below it is not cut and dried that the KSFB is part of the Chew Bahr Rift.

Line 98. As discussed below – there are some clear factors we do know are present that make a clear contrast between the Chew Bahr Rift, and the KSFB (timing of activity, influence of basement fabrics vs presence/absence of the Anza Graben and Pliocene volcanics).

Thank you for pointing this out that carefully. As stated above, we removed the Turkana Region as a natural analogue for our models from the revised manuscript and consequently removed this passage from the introduction.

Lines 114-115. You might also mention there are some studies where stress deflection has been identified in nature (modern stress from boreholes) deflected around faults (e.g., Tingay et al., 2010, JSG).

Thank you for pointing this out. We included the suggested reference as the North Malay Basin provides an excellent and well-studied example of stress deflection and re-orientation near pre-existing faults.

*"… They suggested that pre-existing faults may disturb the local stress field and impede linkage of newly forming faults which also occurs in natural examples of multiphase extension (Duffy et al., 2015). Such stress deflections have been reported and studied in various natural settings such as the North Malay Basin, Thailand, due to the vicinity of pre-existing faults (Tingay et al., 2006; Tingay et al., 2010). …"*

Line 498-499 – well visible sounds like a 'street' term. This is clearly visible, or well-display, or well-developed

We accept the suggested change and adjusted this passage.

511 the threshold for what? Failure?

This refers to the threshold of $10^{-16}$ s$^{-1}$ that we set to distinguish between model locations of active (i.e., ≥ $10^{-16}$ s$^{-1}$) and tectonically inactive domains (i.e., < $10^{-16}$ s$^{-1}$). We adjusted this sentence for clarity.

Line 554. Isotropic areas, into which the rift segments have yet to propagate.

Thank you, we adjusted this sentence.

Line 558 into either

Thank you, we adjusted this part.

Line 686 approximately (replace somewhat).

Thank you, we adjusted this part.

---

## Author Response (AR2)

**Public justification (visible to the public if the article is accepted and published)**:

Dear colleagues,

Thank you for your revised version addressing Guillaume and Chris' comments. Your paper offers a rare combination of analog and numerical experiments on the important process of ongoing stress perturbation in the context of continent rifting. I am confident your paper will be of great interest to our readership.

Before publication, there is however a small issue that needs some consideration. At line 141 you mention that the basal condition delivers a symmetric strain gradient. There is certainly a velocity gradient, however, I wonder if there is a horizontal strain gradient at all. All the foam bars record the same finite strain. Hence, for each bar, dl/lo is the same everywhere at the base of the model, with no strain gradient. If I am correct, I suggest removing at line 141 the reference to strain gradient.

Kind regards,
Patrice

Dear Patrice,

Thank you for your thorough review and helpful suggestions.

Regarding the extension gradient, you are right that our used terminology was unfortunate. We have changed it to "velocity gradient" rather than removing this sentence, since we believe that this is an important aspect of our analog model setup with respect to how we implemented extension velocities in our numerical models. We have changed line 141 as well as the related label in Fig. 2.

Other minor issues:

line 23: Linear or planar?
Linear seems appropriate in our opinion.

line 72: coma after "faults"
That's changed, thank you.

line 78: remove "with time"
That's changed, thank you.

line 90-91: remove "inferred" and change "regionally" by "regional",

That's changed, thank you.

line 106-108: Need some cut and paste: "Such stress deflection due to the vicinity of pre-existing faults have been reported...""

That's changed, thank you.

line 109-110: remove "occurring in continental rifts in 3D"

That's changed, thank you.

line 128: remove "subsequent"

That's changed, thank you.

Line 244: Remove the repetition with line 228-229.

That's changed, thank you.

Line 488: In figure 8, if possible, could you tune down the opacity of the non-defined faulting, or even consider removing it completely? I don't get the strain rate shading. There is a scale covering 1e-19 to 1e-13 per sec with four shading levels, however, there is only one single shading level on the figure (covering 1e-16 to 1e-15 per sec). The topography is very hard to see as well.

We agree, that the colour shading for the strain rate was not well chosen. We have adjusted the strain rate colouring such that the distinction between the steps is clearer. For all strain rates < 1e-16 1/s, values are completely transparent. Also, we removed non-defined $S_{Hmax}$ orientations by setting the opacity = 0. Figure 8 is now easier to read and we applied the same changes to the pertinent supplementary Figures.

Line 503: Coma after "Hereafter".

That's changed, thank you.

Line 756: "palaeo-movements" is a bit vague here. I suggest: "... to infer changes in plate-motion,..."

That's changed, thank you.